# Pluralistic Image Completion with Gaussian Mixture Models

**Xiaobo Xia**[1*]   **Wenhao Yang**[2*]   **Jie Ren**[3]
**Yewen Li**[4]   **Yibing Zhan**[5]   **Bo Han**[6]   **Tongliang Liu**[1†]
[1]TML Lab, University of Sydney   [2]Nanjing University   [3]University of Edinburgh
[4]Nanyang Technological University   [5]JD Explore Academy   [6]Hong Kong Baptist University

## Abstract

Pluralistic image completion focuses on generating both *visually realistic* and *diverse* results for image completion. Prior methods enjoy the empirical successes of this task. However, their used constraints for pluralistic image completion are argued to be *not well interpretable* and *unsatisfactory* from two aspects. First, the constraints for visual reality can be *weakly correlated* to the objective of image completion or even *redundant*. Second, the constraints for diversity are designed to be *task-agnostic*, which causes the constraints to not work well. In this paper, to address the issues, we propose an end-to-end probabilistic method. Specifically, we introduce a unified *probabilistic graph model* that represents the complex interactions in image completion. The entire procedure of image completion is then mathematically divided into several sub-procedures, which helps efficient enforcement of constraints. The sub-procedure directly related to pluralistic results is identified, where the interaction is established by a *Gaussian mixture model* (GMM). The inherent parameters of GMM are *task-related*, which are *optimized adaptively* during training, while the number of its primitives can control the diversity of results conveniently. We formally establish the effectiveness of our method and demonstrate it with comprehensive experiments. The implementation is available at https://github.com/tmllab/PICMM.

## 1   Introduction

Pluralistic image completion refers to the task of filling in the missing region of an incomplete image, so as to produce *visually realistic* and *diverse* image completion solutions [59–61]. Different from single image completion [39, 12, 55, 50] that learns a *deterministic* mapping from an incomplete image to a complete image, and produces a *unique* result, pluralistic image completion can generate *various* results with visually realistic contents. Pluralistic image completion follows the fact that image completion is a *highly subjective* process and benefits a series of applications such as photo restoration [27], object removal [13], and transmission error concealment [37].

Pluralistic image completion is complicated and challenging. The state-of-the-art methods encode images to latent features. To constrain the visual reality of results, *heuristic constraints* are introduced, *e.g.*, the perceptual loss [27] and attention module [59, 44]. To constrain the diversity of results, the distribution of latent features is assumed to be a *unimodal* Gaussian distribution with *predefined parameters* [59]. Image completion is performed by *sampling* from the unimodal Gaussian distribution.

Although the above paradigm enjoys empirical successes of pluralistic image completion, it is *not well interpretable* and *unsatisfactory* in practice. We detail the issues from two aspects. First, the

---

[*]The first two authors made equal contributions.

[†]Corresponding author: Tongliang Liu (tongliang.liu@sydney.edu.au).

36th Conference on Neural Information Processing Systems (NeurIPS 2022).

added constraints for the visual reality of results are based on *general purposes*. These constraints may work well in some tasks. Unfortunately, there is no clear understanding of what role they play in pluralistic image completion. Inappropriate constraints are likely to have side effects. It is not easy to finish the determination of added constraints as our desideratum, especially in complex image completion tasks. Second, the diversity of image completion results is hard to be constrained in reason. For a specific task, the parameters of the unimodal Gaussian distribution are directly related to the diversity. Nevertheless, these parameters are designed to be *prior knowledge* and *task-agnostic*, which cause *unreasonable pluralistic results*.

In this paper, to address the above issues, we present an end-to-end probabilistic method. The method is constructed with a unified *probabilistic graph model* (PGM) [17, 22] and translates the problem of pluralistic image completion into a structured mathematical representation. Specifically, we suppose that pluralistic image completion can be represented with a directed acyclic graph that is designed reasonably. The nodes of the graph denote different kinds of images and corresponding latent features. The edges of the graph represent the interactions between nodes. Based on this directed acyclic graph, the entire procedure of pluralistic image completion is divided into several sub-procedures. Then, we perform different constraints for different sub-procedures as our desideratum.

In particular, the diversity of image completion results is implemented by modeling the interaction between the latent features of the missing region and incomplete image with a *Gaussian mixture model* (GMM) [40, 64].

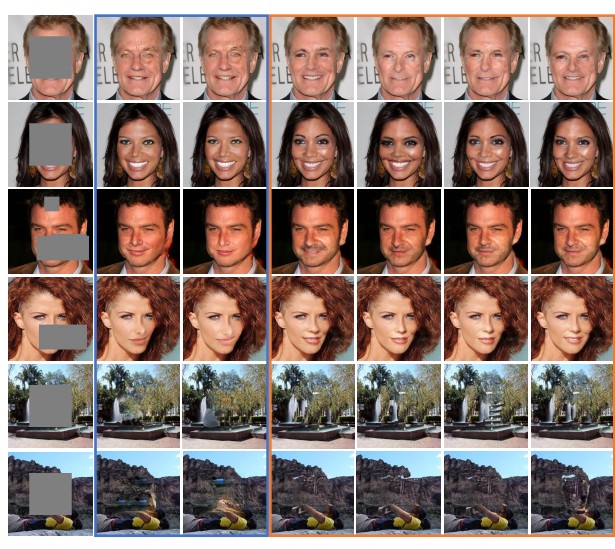

(a) Input      (b) PIC          (c) Ours

Figure 1: **Example completion results of the proposed method on the images of faces and natural sceneries with various masks** (missing regions are shown in gray). The baseline PIC refers to [59]. For each group, the masked input image is shown left, followed by diverse and plausible completion results from our method without any post-processing. (Zoom in to see the details.)

Compared with the unimodal Gaussian distribution, GMM has a larger capacity and is more competitive to meet the output diversity [56, 38]. For our method, the inherent parameters of GMM are *task-related* and *optimized adaptively* during training. The different primitives of GMM represent the outputs with different patterns. Additionally, the number of primitives can be chosen artificially with the needs for diversity in real-world applications.

Before delving into details, we clearly emphasize our contributions as follows:

- To our best knowledge, this paper is the first one that uses a PGM for pluralistic image completion. Besides, the PGM is specially designed with justifications, rather than borrowing existing models. With the designed PGM, the added constraints for visual reality and diversity are explainable.

- We propose to use GMM to increase result diversity. One remarkable advantage is that the inherent parameters for diversity are task-related, leading to more reasonable results.

- We conduct a series of experiments on benchmark datasets to support our claims. In both qualitative and quantitative comparisons with the state-of-the-art methods, our method achieves superior performance. The generated contents for image completion are both visually realistic and diverse, such as those shown in Figure 1.

The rest of the paper is organized as follows. In Section 2, we briefly review pluralistic image completion. In Section 3, we discuss the proposed method step by step. Experimental results are provided in Section 4. Finally, we conclude the paper in Section 5.

## 2 Preliminaries

**Problem Setting.** We first discuss image completion. For the task of image completion, suppose we have an image, originally $\mathbf{I}_o$, but degraded by some missing pixels to become a masked partial image $\mathbf{I}_m$. The image that comprises the original missing pixels is called a complement partial image, which can be denoted by $\mathbf{I}_c$. The goal of the image completion task is to reconstruct

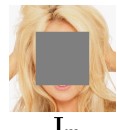 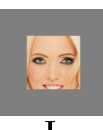 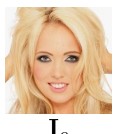
$\mathbf{I}_m$        $\mathbf{I}_c$        $\mathbf{I}_o$

Figure 2: The illustrations for different kinds of images in image completion.

$\mathbf{I}_o$ by using $\mathbf{I}_m$. The illustrations for different kinds of images are provided in Figure 2. Traditional single image completion reconstructs the original image in a *deterministic* fashion, which results in only a single solution [12, 9].

Pluralistic image completion [59] is motivated by the fact that image completion is a highly subjective process. Also, completed images by different experts may agree on high-level semantics, but have substantially different details [59]. Therefore, in pluralistic image completion, we need to generate *multiple* and *diverse* plausible results when presented with a masked partial image. The problem of pluralistic image completion is challenging since we need to take care of the diversity and visual authenticity of results at the same time.

**Prior Work.** Traditional methods on single image completion, *e.g.,* diffusion-based methods [3, 1, 23] and patch-based methods [2, 4, 5], assume that image holes share similar content to visible regions. Therefore, they choose to directly propagate the contextual appearances or realign the background patches to complete the image holes. Benefiting from the strong power of deep neural networks, deep learning based single image completion methods achieve promising completion performance, *e.g.,* employing shepard convolutional neural networks [39], restraining global and local consistency between image holes and visible regions [16], performing deep feature rearrangement [47], and learning a pyramid-context encoder network [53], *etc*. In addition to these methods, multiple single image completion methods involve the generative adversarial networks [10] to learn the semantic of images [48, 49], *e.g.,* [25, 51, 8, 6, 57, 12, 9]. Although these single-solution methods achieve outstanding performance in predicting deterministic result for image holes, they cannot generate various semantically meaningful results.

Existing works for pluralistic image completion usually rely on generative models [24], *e.g.*, CVAE [43, 13, 59], hierarchical VQ-VAE [37], PD-GAN [27], and BicycleGAN [63]. Generative pluralistic image completion methods can diversify meaningful completion results with different kinds of constraints. However, such methods are not well interpretable as discussed. The issues largely limit their applications in the real world.

## 3 Methodology

This section presents a probabilistic method for pluralistic image completion. We first design a probabilistic graph model for our task (Section 3.1). Then, based on the probabilistic graph model, we show how to divide the entire procedure of pluralistic image completion into several sub-procedures (Section 3.2). Afterward, we discuss how to use GMM to diversify image completion results (Section 3.3). Finally, we add reconstruction and adversarial losses to strengthen image completion, and summarize all algorithm flows (Section 3.4).

### 3.1 Probabilistic Graph Model Construction

Recall the notations in image completion, *i.e.*, $\mathbf{I}_o$, $\mathbf{I}_m$, and $\mathbf{I}_c$, they correspond to the latent features $\mathbf{z}_o$, $\mathbf{z}_m$, and $\mathbf{z}_c$ respectively. We suppose a probabilistic graph model for the image completion task, which is shown in Figure 3. The probabilistic graph model is designed reasonably. The graphical illustration of overall computational paths of the probabilistic graph model is presented in Figure 3(d). Specifically, the images $\mathbf{I}_m$ and $\mathbf{I}_c$ form $\mathbf{I}_o$. Their latent features $\mathbf{z}_m$ and $\mathbf{z}_c$ can be exploited to form $\mathbf{I}_o$. Given $\mathbf{z}_m$, we can surmise $\mathbf{z}_c$. We use the latent features here, since they are low-dimensional and informative [11, 21, 45, 46]. We further details Figure 3 as follows.



(a) Inference      (b) Dynamic Inference      (c) Generation      (d) Overall

Figure 3: The illustrations of the probabilistic graph model for the pluralistic image completion task.

**Inference.** We do not directly map $\mathbf{I}_m$ and $\mathbf{I}_c$ to $\mathbf{I}_o$, since they are too high-dimensional and have information redundancy. We exploit generative models. The processes $\mathbf{I}_m \to \mathbf{z}_m$ and $\mathbf{I}_c \to \mathbf{z}_c$ denote that we *encode* images into corresponding latent features. During training, we learn the encoder $f$ to finish the inference.

**Dynamic Inference.** Given $\mathbf{I}_m$ (*resp.* $\mathbf{I}_c$), we can infer the latent feature $\mathbf{z}_m$ (*resp.* $\mathbf{z}_c$). Afterward, $\mathbf{z}_m \dashrightarrow \mathbf{z}_c$ means that we use a variational distribution $p(\mathbf{z}_c|\mathbf{z}_m)$ to approximate the distribution $p(\mathbf{z}_c|\mathbf{I}_c)$. Here we denote this relationship using a dashed line, which is designed to be dynamic to diversify outputs.

**Generation.** In image completion, the images $\mathbf{I}_m$ and $\mathbf{I}_c$ can compose the image $\mathbf{I}_o$. Accordingly, for their latent features, we splice $\mathbf{z}_m$ and $\mathbf{z}_c$ to output $\mathbf{I}_o$. During training, we learn the decoder $g$ to finish the generation.

### 3.2 Image Completion Objective Decomposition

We cannot observe the underlying $\mathbf{I}_c$ in the test procedure. Based on the probabilistic graph model, we perform the following variational inference with the Kullback-Leibler (KL) divergence:

$$L_{\mathrm{P}} = \mathrm{KL}\left[q_\psi(\mathbf{I}_o, \mathbf{z}_m, \mathbf{z}_c|\mathbf{I}_m)\|p_\phi(\mathbf{I}_o, \mathbf{z}_m, \mathbf{z}_c|\mathbf{I}_m, \mathbf{I}_c)\right], \tag{1}$$

where $q_\psi(\cdot|\cdot)$ and $p_\phi(\cdot|\cdot)$ are two conditioned distributions, with $\psi$ and $\phi$ being the parameters of their corresponding functions. The divergence (1) is minimized with respect to all parameters. It should be noted that, in fact, the overall objective should be $\mathrm{KL}[q_\psi(\mathbf{I}_o|\mathbf{I}_m)\|p_\phi(\mathbf{I}_o|\mathbf{I}_m, \mathbf{I}_c)] = \mathrm{KL}[\mathbb{E}_{\mathbf{z}_m,\mathbf{z}_c}[q_\psi(\mathbf{I}_o, \mathbf{z}_m, \mathbf{z}_c|\mathbf{I}_m)]\|\mathbb{E}_{\mathbf{z}_m,\mathbf{z}_c}[p_\phi(\mathbf{I}_o, \mathbf{z}_m, \mathbf{z}_c|\mathbf{I}_m, \mathbf{I}_c)]]$. As the expectation in this formula is *intractable*, we need to deal with the 1-step Monte Carlo sampling objective, which is why the divergence (1) is formed. Another advantage of using the divergence (1) is that constraining each sampling point can make the overall constraint tighter. We further decompose the divergence (1) based on the probabilistic graph model as follows.

**Proposition 1** *Regarding the KL divergence (1), we show that the divergence can be decomposed as:*

$$\begin{aligned} L_{\mathrm{P}} &= \mathrm{KL}\left[q_\psi(\mathbf{I}_o, \mathbf{z}_m, \mathbf{z}_c|\mathbf{I}_m)\|p_\phi(\mathbf{I}_o, \mathbf{z}_m, \mathbf{z}_c|\mathbf{I}_m, \mathbf{I}_c)\right] \\ &= \underbrace{\mathbb{E}_{(\mathbf{z}_m,\mathbf{z}_c)\sim q_\psi(\mathbf{z}_m,\mathbf{z}_c|\mathbf{I}_m)}\mathrm{KL}\left[q_\psi(\mathbf{I}_o|\mathbf{z}_m, \mathbf{z}_c)\|p_\phi(\mathbf{I}_o|\mathbf{I}_m, \mathbf{I}_c)\right]}_{\text{(a)}} \\ &\quad + \underbrace{\mathbb{E}_{\mathbf{z}_m\sim q_\psi(\mathbf{z}_m|\mathbf{I}_m)}\mathrm{KL}\left[q_\theta(\mathbf{z}_c|\mathbf{z}_m)\|p_\phi(\mathbf{z}_c|\mathbf{I}_c)\right]}_{\text{(b)}} + \underbrace{\mathrm{KL}\left[q_\psi(\mathbf{z}_m|\mathbf{I}_m)\|p_\phi(\mathbf{z}_m|\mathbf{I}_m)\right]}_{\text{(c)}}, \end{aligned} \tag{2}$$

*where $q_\theta$ is the variational distribution, and the parameters $\theta$ are involved in divergence minimization.*

The proof can be found in Appendix A.1. We analyze three terms in Eq. (2) combining the probabilistic graph model and our purpose:

- For (a), as discussed, we can generate $\mathbf{I}_o$ with $\mathbf{I}_m$ and $\mathbf{I}_c$, or with their latent features $\mathbf{z}_m$ and $\mathbf{z}_c$. We minimize the distance about the generation of $\mathbf{I}_o$ in the two ways. The minimization can ensure that the generation from the images and latent features is *consistent*.

- For (b), we do not limit the inference from $\mathbf{I}_c$ to $\mathbf{z}_c$ as a deterministic function. Instead, to diversify outputs, we use the variational distribution to restrain $q_\theta(\mathbf{z}_c|\mathbf{z}_m)$. The choice and discussion about the variational distribution will be provided in Section 3.3.

- For ©, it aims to keep that the inference using the variational distribution and the inference using the posterior is *close*, which also guarantees the reliability of sampling. Besides, the variational distribution does not include $\mathbf{I}_c$ to enable latent features to be used in test. We employ the variational evidence lower bound (ELBO) $L_{\text{ELBO}}$ [21, 15] to constrain ©. It is because, suppose that $p(\mathbf{I}_m)$ is a constant given $\mathbf{I}_m$, it is equivalent for maximizing the $L_{\text{ELBO}}$ and minimizing ©. The details are provided in Appendix A.4.

### 3.3 Gaussian Mixture Model

As discussed, we need a suitable choice for the variational distribution $q_\theta(\mathbf{z}_c|\mathbf{z}_m)$ to output pluralistic results. In this work, we employ the Gaussian Mixture Model (GMM) [30, 31] for the dynamic mode $q_\theta(\mathbf{z}_c|\mathbf{z}_m)$, which is more flexible than a unimodal and can diversify outputs.

We denote the latent features of $\mathbf{I}_c$ that are inferred from $\mathbf{z}_m$ by $\hat{\mathbf{z}}_c$, which is distinguished from $\mathbf{z}_c$ achieved by the encoder. For the latent features $\mathbf{z}$, we denote the mean and covariance by $\boldsymbol{\mu}^{\mathbf{z}}$ and $\boldsymbol{\Sigma}^{\mathbf{z}}$. Mathematically, we have

$$q_\theta(\mathbf{z}_c|\mathbf{z}_m) = \sum_{i=1}^{k} \alpha_i \mathcal{N}\left(\mathbf{z}_c|\boldsymbol{\mu}_i^{\hat{\mathbf{z}}_c}, \boldsymbol{\Sigma}_i^{\hat{\mathbf{z}}_c}\right), \tag{3}$$

where $k$ denotes the number of primitives, $\alpha_i$ denotes the mixing coefficient of the $i$-th primitive, and $\boldsymbol{\mu}_i, \boldsymbol{\Sigma}_i$ denotes the distribution parameters of the $i$-th primitive. Note that the mixing coefficient $\alpha_i$ is the parameter of the Categorical distribution [31]. Intuitively, in Eq. (3), we model $\mathbf{z}_m$ with GMM, where the $i$-th primitive can be used to represent $\hat{\mathbf{z}}_c^{(i)}$. In this way, we can obtain different $\hat{\mathbf{z}}_c$ to meet diversity requirements.

Besides, among all the parameters of GMM, except for $k$, the other parameters are called the *inherent parameters* of GMM. In GMM (3), the number of primitives $k$ can be determined artificially according to the need for output diversity. The inherent parameters are optimized adaptively during training. Specifically, for the optimization of the mixing coefficients $\boldsymbol{\alpha} = [\alpha_1, \ldots, \alpha_k]$, we exploit the the *frequency loss* [38]. The objective is

$$L_{\text{F}} = (\mathbf{v} - \boldsymbol{\alpha})(\mathbf{v} - \boldsymbol{\alpha})^\top, \mathbf{v} = [v_1, \ldots, v_k], \text{and}$$
$$v_j = \begin{cases} 1, j = \underset{i}{\operatorname{argmin}} \operatorname{KL}\left[\mathcal{N}(\mathbf{z}_c|\boldsymbol{\mu}_i^{\hat{\mathbf{z}}_c}, \boldsymbol{\Sigma}_i^{\hat{\mathbf{z}}_c})\|\mathcal{N}(\mathbf{z}_c|\boldsymbol{\mu}^{\mathbf{z}_c}, \boldsymbol{\Sigma}^{\mathbf{z}_c})\right]; \\ 0, \text{otherwise.} \end{cases} \tag{4}$$

For our task, with the frequency loss (4), the accumulated frequency approximate gradient is an *asymptotically unbiased estimation* of the true gradient. We provide the detailed derivation and analysis in Appendix A.2. The reason we mention the frequency loss is that, the metric for evaluating the performance of each primitive is adapted to KL divergence in our task, which is different from the original metric in [38]. For the optimization of $\boldsymbol{\mu}$ and $\boldsymbol{\Sigma}$ of $\mathbf{z}_m$, we utilize the *back-propogate-max-operation* to reserve the distinguishable property of GMM (3). For our problem, the objective of the back-propogate-max-operation is shown in the following proposition.

**Proposition 2** *By modeling the variational distribution with GMM (3), we have*

$$L_{\text{BM}} = -\frac{1}{2}\log\frac{|\boldsymbol{\Sigma}_j^{\hat{\mathbf{z}}_c}|}{|\boldsymbol{\Sigma}^{\mathbf{z}_c}|} + \frac{1}{2}\operatorname{tr}\left((\boldsymbol{\Sigma}^{\mathbf{z}_c})^{-1}\boldsymbol{\Sigma}_j^{\hat{\mathbf{z}}_c}\right) - \frac{1}{2}(\boldsymbol{\mu}_j^{\hat{\mathbf{z}}_c} - \boldsymbol{\mu}^{\mathbf{z}_c})^\top(\boldsymbol{\Sigma}^{\mathbf{z}_c})^{-1}(\boldsymbol{\mu}_j^{\hat{\mathbf{z}}_c} - \boldsymbol{\mu}^{\mathbf{z}_c}), \tag{5}$$

where $j = \operatorname{argmin}_i \operatorname{KL}\left[\mathcal{N}(\mathbf{z}_c|\boldsymbol{\mu}_i^{\hat{\mathbf{z}}_c}, \boldsymbol{\Sigma}_i^{\hat{\mathbf{z}}_c})\|\mathcal{N}(\mathbf{z}_c|\boldsymbol{\mu}^{\mathbf{z}_c}, \boldsymbol{\Sigma}^{\mathbf{z}_c})\right]$, and $\operatorname{tr}(\cdot)$ denotes the trace of a matrix. The derivation of Proposition 2 can be found in Appendix A.3. For Proposition 2, if we minimize $L_{\text{BM}}$, we optimize $\boldsymbol{\mu}$ and $\boldsymbol{\Sigma}$ of $\mathbf{z}_m$, which reduces the distance between $\hat{\mathbf{z}}_c^{(j)}$ and $\mathbf{z}_c$. For the term ⓑ in Eq. (2), we then have $L_{\text{GMM}} = \mathbb{E}_{\mathbf{z}_m \sim q_\psi(\mathbf{z}_m|\mathbf{I}_m)}\operatorname{KL}\left[q_\theta(\mathbf{z}_c|\mathbf{z}_m)\|p_\phi(\mathbf{z}_c|\mathbf{I}_c)\right] \approx L_{\text{F}} + L_{\text{BM}}$.

**Discussion.** Note that the diversity of results is directly related to $\mathbf{z}_m$. When GMM for $\mathbf{z}_m$ is *dominated* by the $i$-th primitive ($\alpha_i \approx 1$), the diversity of image completion results is reduced. Although this phenomenon did not appear in our experiments, it may exist in practice. Actually, this phenomenon is reasonable. From human cognition, if $\mathbf{I}_m$ shows *only one pattern*, the inference for $\mathbf{I}_c$ will be deterministic [3]. For example, if the size of the missing region of an incomplete image is

very small, we can infer the missing region both deterministically and reasonably, where the diversity of results is reduced. Prior methods with task-agnostic constraints do not consider this fact, which may create *unreasonable results*. For our method, the constraints for diversity are task-related, which can take this phenomenon into account.

---

**Algorithm 1** Training procedure

**Input:** images $\mathbf{I}_o$, $\mathbf{I}_m$, and $\mathbf{I}_c$, the number of primitives of GMM $k$, the initialized encoder $f$, and decoder $g$.

1: **Encode** $\mathbf{I}_m$ (*resp.* $\mathbf{I}_c$) to $\mathbf{z}_m$ (*resp.* $\mathbf{z}_c$) with $f$;
2: **Model** $\mathbf{z}_m$ with GMM and $\mathbf{z}_c$ with a unimodal Gaussian distribution;
3: **Calculate** the loss $L_{\text{GMM}}$ as discussed in Section 3.3;
4: **Update** the parameters of GMM with $L_{\text{GMM}}$;
5: **Infer** $\hat{\mathbf{z}}_c^{(j)}$ from $\mathbf{z}_m$ with Eq. (3) and Eq. (4), $j = 1, \ldots, k$;
6: **Generate** images by $g$ with $\hat{\mathbf{I}}_o = g(\mathbf{z}_m, \mathbf{z}_c)$ and $\hat{\mathbf{I}}_o^{(j)} = g(\mathbf{z}_m, \hat{\mathbf{z}}_c^{(j)})$, $j = 1, \ldots, k$;
7: **Calculate** the loss $L_{\text{ELBO}}$ and $L_{\text{C}}$ as discussed in Section 3.2 and 3.4;
8: **Update** all the parameters *w.r.t.* $L_{\text{ELBO}}$ and $L_{\text{C}}$.

**Output:** the trained encoder $f^*$ and decoder $g^*$.

---

**Algorithm 2** Test procedure

**Input:** the image $\mathbf{I}_m$, the number of primitives of GMM $k$, the trained encoder $f^*$, and decoder $g^*$.

1: **Encode** $\mathbf{I}_m$ to $\mathbf{z}_m$ with $f^*$;
2: **Model** $\mathbf{z}_m$ with GMM;
3: **for** $j = 1$ to $k$ **do**
4:     **Sample** $i$ with $i \sim$ Categorical$(\alpha_1, \ldots, \alpha_k)$;
5:     **Infer** $\hat{\mathbf{z}}_c^{(j)}$ with $\hat{\mathbf{z}}_c^{(j)} = \mathcal{N}(\boldsymbol{\mu}_i^{\mathbf{z}_m}, \boldsymbol{\Sigma}_i^{\mathbf{z}_m})$;
6:     **Generate** images $\hat{\mathbf{I}}_o^{(j)}$ by $g^*$ with $\hat{\mathbf{I}}_o^{(j)} = g^*(\mathbf{z}_m, \hat{\mathbf{z}}_c^{(j)})$.
7: **end for**

**Output:** pluralistic image completion results $\{\hat{\mathbf{I}}_o^{(j)}\}_{j=1}^k$.

---

## 3.4 Reconstruction and Adversarial Losses

To ensure the minimization of the term ⓐ in Eq. (2), we incorporate the use of reconstruction and adversarial losses [10]. The reconstruction loss is formulated as:

$$L_{\text{R}} = \mathbb{E}\left[\|\hat{\mathbf{I}}_o - \mathbf{I}_o\|_1\right] + \mathbb{E}\left[\|\hat{\mathbf{I}}_m^{(j)} - \mathbf{I}_m\|_1\right], \tag{6}$$

where $\hat{\mathbf{I}}_o$ is generated from $\mathbf{z}_m$ and $\mathbf{z}_c$, and $\hat{\mathbf{I}}_m^{(j)}$ is generated from $\hat{\mathbf{z}}_c^{(j)}$. Here, $\hat{\mathbf{z}}_c^{(j)}$ can be inferred from $\mathbf{z}_m$ by using Eq. (3) and Eq. (4). The image $\hat{\mathbf{I}}_m^{(j)}$ is obtained from $\hat{\mathbf{I}}_o^{(j)}$ with the image mask that can degenerate $\mathbf{I}_o$ to $\mathbf{I}_m$. The image $\hat{\mathbf{I}}_o^{(j)}$ is generated from $\mathbf{z}_m$ and $\hat{\mathbf{z}}_c^{(j)}$. The reconstruction loss (6) controls the visual rationality of image completion results. Also, we efficiently avoid the deterministic fashion with GMM. Furthermore, we leverage adversarial training [10] to make the generated images more realistic. The adversarial loss $L_{\text{A}}$ is formulated as

$$L_{\text{A}} = \mathbb{E}\left[\|\mathcal{D}(\hat{\mathbf{I}}_o) - 1\|_2^2\right] + \mathbb{E}\left[\|\mathcal{D}(\hat{\mathbf{I}}_o^{(j)}) - \mathcal{D}(\mathbf{I}_o)\|_2^2\right], \tag{7}$$

where $\mathcal{D}$ is the discriminator optimized by the discriminator loss based on LSGAN [29]. We jointly train our encoder $f$ and decoder $g$ through the following combined loss:

$$L_{\text{C}} = L_{\text{R}} + \lambda_{\text{A}} L_{\text{A}}, \tag{8}$$

where the weight $\lambda_{\text{A}}$ is set to 0.05 in all experiments. Then, the final objective of the loss function is the sum of three losses, *i.e.*, $L_{\text{FINAL}} = L_{\text{GMM}} + L_{\text{ELBO}} + L_{\text{C}}$. The final loss is to maximize the log-likelihood of the conditional data distribution. Specifically, the adversarial loss uses the Wasserstein distance to make generated images realistic given latent features. The ELBO loss uses KL divergence and distribution log-likelihood to make variational posterior approximate real posterior.

**Algorithm Flows.** For the convenience of following technical details, we provide the algorithm flows. The algorithm flows of training and test stages can be found in Algorithm 1 and Algorithm 2. The illustration is shown in Appendix B.

It should be noted that the number of primitives $k$ does not mean that we are limited to generate only $k$ diverse results given $\mathbf{I}_m$ (in Steps 3 and 4 of Algorithm 2). In fact, we can sample from the $k$ primitives to obtain lots of image completion results. Moreover, as GMM has a much larger capacity than the unimodal Gaussian distribution [31], we can achieve greater diversity. We provide empirical observations in Section 4.3 and Appendix C.

## 4 Experiments

In this section, we conduct a series of experiments to justify our claims. We first introduce the implementation of our method (Section 4.1). The comprehensive experimental results and comparison with advanced methods are then provided and discussed (Section 4.2). Finally, we conduct an analysis study to present and discuss our method in more detail (Section 4.3).

### 4.1 Implementation Details

**Datasets.** We evaluated our proposed model on five popularly used datasets, *i.e.*, CelebA-HQ [18, 28], FFHQ [19], Paris StreetView [7], Places2 [62], and ImageNet [41]. We verify the effectivness of our method with different types of mask regions, including both center and random masks [59, 44].

**Network and Optimization.** The proposed method can be implemented efficiently. The encoder and decoder of our pipeline are inspired by PIC [59] for comparison. We apply average pooling and interpolation with convolutional layers to implement downsampling and upsampling respectively. During optimization, we use the Adam optimizer [20]. The learning rate is fixed to $10^{-4}$ during the training procedure.

**Baselines.** We compare the proposed method with the following state-of-the-art methods, which include (1) single image completion methods: DFv2 [52], EC [33], and MED [26]; (2) pluralistic image completion methods: PIC [59] and ICT [44]. We abbreviate our method (**P**luralistic **I**mage **C**ompletion with Gaussian **M**ixture **M**odels) as PICMM. The methods are implemented by PyTorch and evaluated on NVIDIA Tesla A100 GPUs.

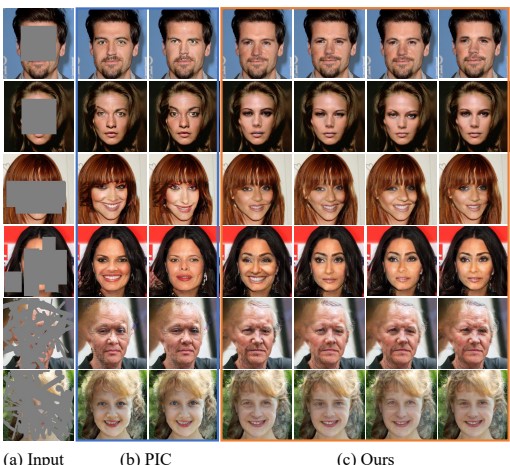

(a) Input      (b) PIC      (c) Ours

Figure 4: Qualitative comparison of our method with PIC on CelebA-HQ (**first 4 rows**) and FFHQ (**last 2 rows**). Best viewed by zooming in.

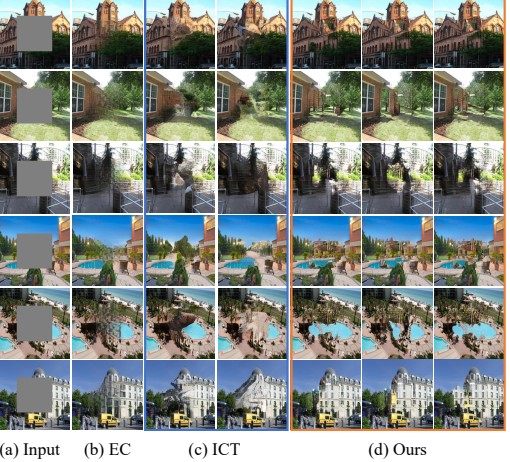

(a) Input   (b) EC   (c) ICT   (d) Ours

Figure 6: Qualitative comparison of our method with EC and ICT on Places2. Best viewed by zooming in.

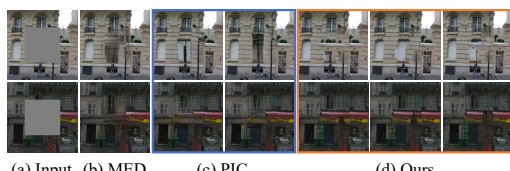

(a) Input   (b) MED   (c) PIC      (d) Ours

Figure 5: Qualitative comparison of our method with MED and PIC on Paris StreetView. Best viewed by zooming in.

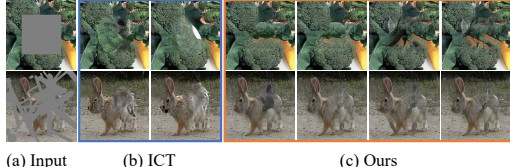

(a) Input      (b) ICT      (c) Ours

Figure 7: Qualitative comparison of our method with ICT on ImageNet. Best viewed by zooming in.

**Measurement.** For measurement, we provide both qualitative and quantitative results. For the metrics of quantitative results, we use common evaluation metrics such as the peak signal-to-noise

ratio (PSNR), structural similarity (SSIM), mean absolute error (MAE), and fréchet inception distance (FID) [14] to measure the similarity between the image completion result and ground truth. Furthermore, pluralistic image completion is supposed to focus on generating diverse realistic results rather than merely approximating ground-truth ones [58]. To *measure the result diversity*, we add two perceptual quality metrics to quantitative results, which include LPIPS [59] and DIV-FID [44].

## 4.2 Comparison with Prior Methods

### 4.2.1 Qualitative Comparisons

We provide extensive qualitative comparison results to justify our claims. First, we show the results on CelebA-HQ and FFHQ in Figure 4, which are images related to human faces. As can be seen, the image completion results achieved by our method are realistic, following high image quality. Also, compared with the baseline PIC, our completion results are more diverse, *e.g.*, see the images in the second and last rows.

Second, we provide the results on Paris StreetView and Places in Figures 5 and 6 respectively. For the results on Paris StreetView, we can see that image completion by the baselines MED and PIC are somewhat distorted. By contrast, our method can finish the image completion task better. The details can be checked in the complements to walls. Then, we turn the attention to the image completion results on Places2. Our method still achieves superior performance compared with baselines.

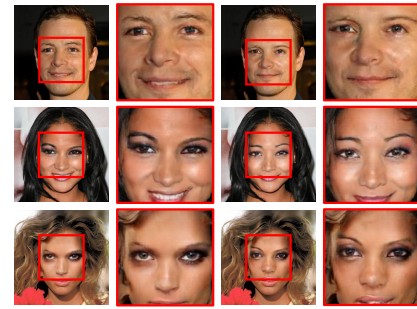

Figure 8: Refined result analysis of our method. The face images come from CelebA-HQ.

Lastly, we provide the results on ImageNet in Figure 7. Note that the sources of the data in ImageNet are very complicated. The image completion task is rather challenging on this dataset, even for single image completion [59]. Different from some prior work [36, 16] that were trained on a 100k subset of training images of ImageNet, we directly the network on the original ImageNet training dataset with all images. The results mean that our method can infer the content effectively.

**Refined Result Analysis.** We provide some refined results in Figure 8 to analyze the image completion details of our method. For the face images, we can find that the image completion results have very different facial expressions.

| Dataset | | FFHQ | | | | Places2 | | | |
|---|---|---|---|---|---|---|---|---|---|
| Method | Mask | PSNR ↑ | SSIM ↑ | MAE ↓ | FID ↓ | PSNR ↑ | SSIM ↑ | MAE ↓ | FID ↓ |
| DFv2 [52] | Center | 25.868 | 0.922 | 0.0231 | 16.278 | 26.533 | 0.881 | 0.0215 | 24.763 |
| EC [33] | | 26.901 | 0.938 | 0.0209 | 14.276 | 26.520 | 0.880 | 0.0220 | 25.642 |
| MED [26] | | 26.325 | 0.922 | 0.0230 | 14.791 | 26.469 | 0.877 | 0.0224 | 26.977 |
| PIC [59] | | 26.781 | 0.933 | 0.0215 | 14.513 | 26.099 | 0.865 | 0.0236 | 26.393 |
| ICT [44] | | **27.922** | **0.948** | 0.0208 | 10.995 | 26.503 | 0.880 | 0.0244 | **21.598** |
| PICMM† | | 27.551 | 0.937 | **0.0203** | 10.604 | **26.554** | **0.886** | **0.0208** | 24.373 |
| DFv2 [52] | Random | 24.962 | 0.882 | 0.0310 | 19.506 | 25.692 | **0.834** | 0.0280 | 29.981 |
| EC [33] | | 25.908 | 0.882 | 0.0301 | 17.039 | 25.510 | 0.831 | 0.0293 | 30.130 |
| MED [26] | | 25.118 | 0.867 | 0.0349 | 19.644 | 25.632 | 0.827 | 0.0291 | 31.395 |
| PIC [59] | | 25.580 | 0.889 | 0.0303 | 17.364 | 25.035 | 0.806 | 0.0315 | 33.472 |
| ICT [44] | | **26.681** | 0.910 | 0.0292 | **14.529** | **25.788** | 0.832 | 0.0267 | 25.420 |
| PICMM† | | 25.950 | **0.912** | **0.0289** | 17.014 | 25.310 | 0.829 | **0.0236** | **25.025** |

Table 1: Quantitative results on FFHQ and Places2 datasets with different mask settings. The best results are in **bold**.

### 4.2.2 Quantitative Comparisons

**Common Metrics.** Quantitative evaluation with common metrics is difficult for pluralistic image completion, as our goal is to obtain diverse but reasonable solutions for a masked image [59]. To make a feasible comparison, as did in [59], we sample one completed image many times from $k$ images to compare with the original image. The results are provided in Tables 1 and 2. As can be seen, for FFHQ and Places2, on both the set-

| Dataset | | ImageNet | | | |
|---|---|---|---|---|---|
| Method | Mask | PSNR ↑ | SSIM ↑ | MAE ↓ | FID ↓ |
| PIC [59] | | 24.010 | 0.867 | 0.0319 | 47.750 |
| ICT [44] | Center | 24.757 | 0.888 | 0.0263 | 28.818 |
| PICMM† | | **24.932** | **0.897** | **0.0260** | **23.718** |
| PIC [59] | | 22.711 | 0.791 | 0.0462 | 59.428 |
| ICT [44] | Random | **23.775** | 0.835 | **0.0358** | **35.842** |
| PICMM† | | 23.322 | **0.846** | 0.0415 | 39.742 |

Table 2: Quantitative results on ImageNet with different mask settings. The best results are in **bold**.

tings of center and random masking, our method achieves superior performance. On ImageNet, our method performs the best with the center masking consistently, while it is competitive with the random masking.

It should be noted that ICT is a very strong baseline and works better than our method in some cases. It is because ICT is built with the image transformer [35], which is powerful in vision tasks. Compared with ICT, the advantages of our method lies in not only better interpretability as discussed, but also faster inference speeds. We will demonstrate this in inference time.

**Diversity Metrics.** For pluralistic image completion, the diversity evaluations are significant. We generate five completion images in each task and report the mean of the diversity metric. The results are shown in Table 3, which represents that our method can generate more diverse completion images than baselines. The results with common and diversity metrics support

| Dataset | FFHQ | | Places2 | |
|---|---|---|---|---|
| Method | LPIPS ↑ | DIV-FID ↑ | LPIPS ↑ | DIV-FID ↑ |
| PIC [59] | 0.029 | 9.130 | 0.047 | 17.742 |
| ICT [44] | 0.065 | 13.909 | 0.089 | 25.253 |
| PICMM† | **0.071** | **17.361** | **0.092** | **27.194** |

Table 3: The diversity result comparison on FFHQ and Places2 datasets with the center mask setting. The best results are in **bold**.

our claim very well. That is to say, the image completion results of our method are both high-quality and pluralistic.

**Inference Time.** We report the inference time of our method, compared with PIC and ICT. All inference runs on one NVIDIA Tesla A100 GPU for fairness. The size of images is $256 \times 256$. We perform pluralistic image completion for 100 different images. For each image, we generate 6 completion results for it. We report the total inference time for these images. The results are shown in Table 4. Note that as ICT is a transformer-based method, the calculated consumption of its infer-

| Method | Inference time (minutes) ↓ |
|---|---|
| PIC [59] | 0.387 |
| ICT [44] | 179.126 |
| PICMM† | **0.235** |

Table 4: The inference time comparison with PIC and ICT in the center mask setting. The best result is in **bold**.

ence is much heavier. What's worse, ICT needs *iterative Gibbs sampling* during inference. The two issues make ICT have to face heavy computational consumption, which is also mentioned in [44]. In contrast, our method is more inference-efficiency, which is **more than 750 times faster than ICT**. The merit could make our method easier to use in the real world.

### 4.3 More Analyses and Justifications

**GMM Primitive Visualization.** We further stress the diversity of the completion results achieved by our method. Specifically, we visualize the $\hat{\mathbf{z}}_c$ which is inferred from $\mathbf{z}_m$ as discussed. The visualization results are obtained with t-SNE [42]. Different $\hat{\mathbf{z}}_c$ are presented by the 2D-vectors in different colors, which are shown in Figure 9. As can be seen, the vectors are *scattered*, which clearly demonstrates that the completion results of our method are diverse.

**Refined Diversity Comparison.** We argue that, benefiting from that GMM has larger capacities than a unimodal Gaussian distribution, the completion results of our method would be more diverse. To justify our claims, we provide the refined diversity comparison of our method with PIC and ICT, as shown in Figure 10. Note that the six image completion results of PIC are obtained by sampling from

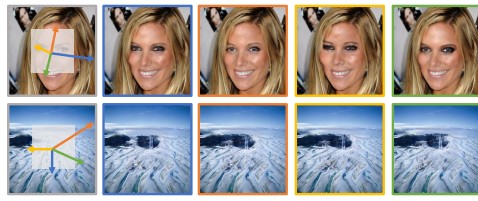

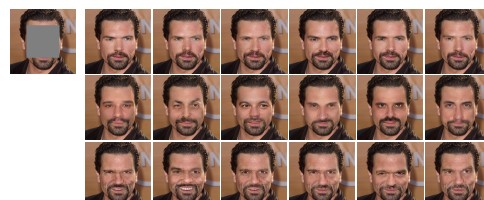

Figure 9: GMM primitive visualization of pluralistic completion results by our method. The face image comes from CelebA-HQ. The scene image comes from Places2.

Figure 10: Refined diversity comparison with PIC (**the first row**) and ICT (**the second row**). The original image in this figure comes from CelebA-HQ.

one unimodal Gaussian distribution. The results of our method are obtained by sampling from six primitives. Clearly, our results are more diverse than PIC's. Besides, for ICT, it does not explicitly control the output diversity. The diversity of pluralistic image completion is thus *less interpretable* than our method.

In addition, we state that our method is not limited to only generating $k$ completion results before. In fact, we can sample from $k$ primitives of GMM to obtain lots of completion results. Due to the limited page, we present the results and more analyses in Appendix C. Moreover, the richer completion result comparison is presented in Appendix D.

## 5    Conclusion

In this paper, we focus on the complicated and challenging problem of pluralistic image completion. We propose a novel end-to-end probabilistic method for this challenging problem. Based on the probabilistic graph model, our method divides the entire procedure of pluralistic image completion into several sub-procedures, where GMM is used to diversify outputs. Experiments on a variety of datasets show that the image completion results by our method are both high-quality and pluralistic. In future work, investigating the feasibility of the proposed method for other diverse decision-making scenarios might prove important.

## Acknowledgements

Yibing Zhan was partially supported by the Major Science and Technology Innovation 2030 "New Generation Artificial Intelligence" key project (No. 2021ZD0111700) and the National Natural Science Foundation of China (Grant No. 62002090). Xiaobo Xia was supported by Australian Research Council Projects DE-190101473 and Google PhD Fellowship. Bo Han was supported by the RGC Early Career Scheme No. 22200720, NSFC Young Scientists Fund No. 62006202, and Guangdong Basic and Applied Basic Research Foundation No. 2022A1515011652. Tongliang Liu was partially supported by Australian Research Council Projects DP180103424, DE-190101473, IC-190100031, DP-220102121, and FT-220100318. The authors would give special thanks to Mingrui Zhu (Xidian University), Zihan Ding (Princeton University), and Chenlai Qian (Southeast University) for helpful discussions.

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
