# Supplementary Material of "Pluralistic Image Completion with Gaussian Mixture Models"

The appendix is organized as follows. We detail theoretical analyses in Section A. The related work is reviewed in Section B. Supplementary experimental results are provided in Section C and Section D.

## A  The Details of Theoretical Analyses

### A.1  The Proof of Proposition 1

As we do not have the access to the underlying $\mathbf{I}_c$, we can preform the following variational approximation with the Kullback-Leibler (KL) divergence:

$$
\begin{aligned}
\min \quad & \mathrm{KL}\left[q_\psi(\mathbf{I}_o, \mathbf{z}_m, \mathbf{z}_c | \mathbf{I}_m) \| p_\phi(\mathbf{I}_o, \mathbf{z}_m, \mathbf{z}_c | \mathbf{I}_m, \mathbf{I}_c)\right] \\
= & \int_{\mathbf{I}_o} \int_{\mathbf{z}_m} \int_{\mathbf{z}_c} q_\psi(\mathbf{I}_o, \mathbf{z}_m, \mathbf{z}_c | \mathbf{I}_m) \log \frac{q_\psi(\mathbf{I}_o, \mathbf{z}_m, \mathbf{z}_c | \mathbf{I}_m)}{p_\phi(\mathbf{I}_o, \mathbf{z}_m, \mathbf{z}_c | \mathbf{I}_m, \mathbf{I}_c)} \mathrm{d}\mathbf{I}_o \mathrm{d}\mathbf{z}_m \mathrm{d}\mathbf{z}_c \\
= & \int_{\mathbf{I}_o} \int_{\mathbf{z}_m} \int_{\mathbf{z}_c} q_\psi(\mathbf{I}_o | \mathbf{z}_m, \mathbf{z}_c) q_\psi(\mathbf{z}_m, \mathbf{z}_c | \mathbf{I}_m) \log \frac{q_\psi(\mathbf{I}_o | \mathbf{z}_m, \mathbf{z}_c) q_\psi(\mathbf{z}_m, \mathbf{z}_c | \mathbf{I}_m)}{p_\phi(\mathbf{I}_o | \mathbf{I}_m, \mathbf{I}_c) p_\phi(\mathbf{z}_m, \mathbf{z}_c | \mathbf{I}_m, \mathbf{I}_c)} \mathrm{d}\mathbf{I}_o \mathrm{d}\mathbf{z}_m \mathrm{d}\mathbf{z}_c \\
= & \int_{\mathbf{I}_o} \int_{\mathbf{z}_m} \int_{\mathbf{z}_c} q_\psi(\mathbf{I}_o | \mathbf{z}_m, \mathbf{z}_c) q_\psi(\mathbf{z}_m, \mathbf{z}_c | \mathbf{I}_m) \log \frac{q_\psi(\mathbf{I}_o | \mathbf{z}_m, \mathbf{z}_c)}{p_\phi(\mathbf{I}_o | \mathbf{I}_m, \mathbf{I}_c)} \mathrm{d}\mathbf{I}_o \mathrm{d}\mathbf{z}_m \mathrm{d}\mathbf{z}_c \\
& + \int_{\mathbf{I}_o} \int_{\mathbf{z}_m} \int_{\mathbf{z}_c} q_\psi(\mathbf{I}_o | \mathbf{z}_m, \mathbf{z}_c) q_\psi(\mathbf{z}_m, \mathbf{z}_c | \mathbf{I}_m) \log \frac{q_\psi(\mathbf{z}_m, \mathbf{z}_c | \mathbf{I}_m)}{p_\phi(\mathbf{z}_m, \mathbf{z}_c | \mathbf{I}_m, \mathbf{I}_c)} \mathrm{d}\mathbf{I}_o \mathrm{d}\mathbf{z}_m \mathrm{d}\mathbf{z}_c \\
= & \int_{\mathbf{z}_m} \int_{\mathbf{z}_c} q_\psi(\mathbf{z}_m, \mathbf{z}_c | \mathbf{I}_m) \mathrm{KL}\left[q_\psi(\mathbf{I}_o | \mathbf{z}_m, \mathbf{z}_c) \| p_\phi(\mathbf{I}_o | \mathbf{I}_m, \mathbf{I}_c)\right] \mathrm{d}\mathbf{z}_m \mathrm{d}\mathbf{z}_c \\
& + \int_{\mathbf{I}_o} q_\psi(\mathbf{I}_o | \mathbf{z}_m, \mathbf{z}_c) \mathrm{KL}\left[q_\psi(\mathbf{z}_m, \mathbf{z}_c | \mathbf{I}_m) \| p_\phi(\mathbf{z}_m, \mathbf{z}_c | \mathbf{I}_m, \mathbf{I}_c)\right] \mathrm{d}\mathbf{I}_o \\
= & \mathbb{E}_{(\mathbf{z}_m, \mathbf{z}_c) \sim q_\psi(\mathbf{z}_m, \mathbf{z}_c | \mathbf{I}_m)} \mathrm{KL}\left[q_\psi(\mathbf{I}_o | \mathbf{z}_m, \mathbf{z}_c) \| p_\phi(\mathbf{I}_o | \mathbf{I}_m, \mathbf{I}_c)\right] + \mathrm{KL}\left[q_\psi(\mathbf{z}_m, \mathbf{z}_c | \mathbf{I}_m) \| p_\phi(\mathbf{z}_m, \mathbf{z}_c | \mathbf{I}_m, \mathbf{I}_c)\right].
\end{aligned}
\tag{9}
$$

The second term in Eq. (9) is

$$
\begin{aligned}
& \mathrm{KL}\left[q_\psi(\mathbf{z}_m, \mathbf{z}_c | \mathbf{I}_m) \| p_\phi(\mathbf{z}_m, \mathbf{z}_c | \mathbf{I}_m, \mathbf{I}_c)\right] \\
= & \int_{\mathbf{z}_m} \int_{\mathbf{z}_c} q_\theta(\mathbf{z}_c | \mathbf{z}_m) q_\psi(\mathbf{z}_m | \mathbf{I}_m) \log \frac{q_\theta(\mathbf{z}_c | \mathbf{z}_m) q_\psi(\mathbf{z}_m | \mathbf{I}_m)}{p_\phi(\mathbf{z}_m, \mathbf{z}_c | \mathbf{I}_m, \mathbf{I}_c)} \mathrm{d}\mathbf{z}_m \mathrm{d}\mathbf{z}_c \\
= & \int_{\mathbf{z}_m} \int_{\mathbf{z}_c} q_\theta(\mathbf{z}_c | \mathbf{z}_m) q_\psi(\mathbf{z}_m | \mathbf{I}_m) \log \frac{q_\theta(\mathbf{z}_c | \mathbf{z}_m)}{p_\phi(\mathbf{z}_c | \mathbf{I}_c)} \mathrm{d}\mathbf{z}_m \mathrm{d}\mathbf{z}_c + q_\theta(\mathbf{z}_c | \mathbf{z}_m) q_\psi(\mathbf{z}_m | \mathbf{I}_m) \log \frac{q_\psi(\mathbf{z}_m | \mathbf{I}_m)}{p_\phi(\mathbf{z}_m | \mathbf{I}_m)} \mathrm{d}\mathbf{z}_m \mathrm{d}\mathbf{z}_c \\
= & \mathbb{E}_{\mathbf{z}_m \sim q_\psi(\mathbf{z}_m | \mathbf{I}_m)} \mathrm{KL}\left[q_\theta(\mathbf{z}_c | \mathbf{z}_m) \| p_\phi(\mathbf{z}_c | \mathbf{I}_c)\right] + \mathrm{KL}\left[q_\psi(\mathbf{z}_m | \mathbf{I}_m) \| p_\phi(\mathbf{z}_m | \mathbf{I}_m)\right].
\end{aligned}
\tag{10}
$$

Combining Eq. (9) and Eq. (10), we complete the proof. ∎

Note that the latent variables $\mathbf{z}_m$ and $\mathbf{z}_c$ are *not codependent*. If $\mathbf{z}_m$ and $\mathbf{z}_c$ are codependent, the inference of $\mathbf{z}_m$ (*resp.* $\mathbf{z}_c$) *must need* $\mathbf{z}_c$ (*resp.* $\mathbf{z}_m$). Then, in Eq. (10), there should be $q_\psi(\mathbf{z}_m | \mathbf{I}_m, \mathbf{z}_c)$ and $p_\phi(\mathbf{z}_c | \mathbf{I}_c, \mathbf{z}_m)$, rather than $q_\psi(\mathbf{z}_m | \mathbf{I}_m)$ and $p_\phi(\mathbf{z}_c | \mathbf{I}_c)$. As shown in the our Figure 3, $\mathbf{z}_m$ and $\mathbf{z}_c$ can be inferred without each other, but using $\mathbf{I}_m$ and $\mathbf{I}_c$. Therefore, $q_\psi(\mathbf{z}_m | \mathbf{I}_m)$ and $p_\phi(\mathbf{z}_c | \mathbf{I}_c)$ hold in Eq. (10).

### A.2  The Derivation of the Frequency Loss

As discussed, we exploit the frequency loss [38] for the Gaussian mixture model. Here, we detail The derivation of the frequency loss for our task, *i.e.*, pluralistic image completion. Specifically, we use the KL divergence between each primitive $\mathcal{N}(\mathbf{z}_c | \boldsymbol{\mu}_i^{\hat{\mathbf{z}}_c}, \boldsymbol{\Sigma}_i^{\hat{\mathbf{z}}_c})$ and $\mathcal{N}(\mathbf{z}_c | \boldsymbol{\mu}^{\mathbf{z}_c}, \boldsymbol{\Sigma}^{\mathbf{z}_c})$ as the metric of the performance:

$$
\mathrm{KL}\left[\mathcal{N}(\mathbf{z}_c | \boldsymbol{\mu}_i^{\hat{\mathbf{z}}_c}, \boldsymbol{\Sigma}_i^{\hat{\mathbf{z}}_c}) \| \mathcal{N}(\mathbf{z}_c | \boldsymbol{\mu}^{\mathbf{z}_c}, \boldsymbol{\Sigma}^{\mathbf{z}_c})\right], i = 1, ..., k.
\tag{11}
$$

Then, by comparing the KL divergence of each primitive, we can get the best primitive's index $j$. That is

$$j = \underset{i}{\arg\min} \, \text{KL}\left[\mathcal{N}(\mathbf{z}_c|\boldsymbol{\mu}_i^{\hat{\mathbf{z}}_c}, \boldsymbol{\Sigma}_i^{\hat{\mathbf{z}}_c}) \| \mathcal{N}(\mathbf{z}_c|\boldsymbol{\mu}^{\mathbf{z}_c}, \boldsymbol{\Sigma}^{\mathbf{z}_c})\right]. \tag{12}$$

We now can give a detailed derivation for Eq. (4) below. Specifically, for stochastic Gaussian mixture models, the gradient value of a single-instance sampling process from each primitive's performance under the metric Eq. (11), to parameters $\theta_{\boldsymbol{\alpha}}$ that output the mixing coefficients $\boldsymbol{\alpha} = [\alpha_1, \ldots, \alpha_k]$, can be estimated with a *frequency approximate gradient*. Mathematically, we have

$$\text{grad}_i = \delta_i \nabla_{\theta_i} \alpha_i \text{ and } \delta_i = -\mathbb{1}_i^{\text{best}} + \alpha_i, \tag{13}$$

where $\delta_i$ is the gradient of the performance metric Eq. (11) for $\alpha_i$, $\nabla_{\theta_i} \alpha_i$ is the gradient of $\alpha_i$ for parameters $\theta_i$, and $\mathbb{1}_i^{\text{best}}$ is the indicator function, where $\mathbb{1}_i^{\text{best}} = 1$ if $i = j$ and $\mathbb{1}_i^{\text{best}} = 0$ otherwise.

The accumulated frequency approximate gradient is an *asymptotically unbiased estimation* of the true gradient for the sampling process from a categorical distribution $\boldsymbol{\alpha}$, with a batch of $n \to \infty$ examples. For $\delta_i$, we can detail it below:

$$\begin{aligned}
\delta_i &= -\mathbb{1}_i^{\text{best}} + \alpha_i \\
&= \mathbb{1}_i^{\text{best}} \alpha_i - \mathbb{1}_i^{\text{best}} \alpha_i - \mathbb{1}_i^{\text{best}} + \alpha_i \\
&= \frac{1}{2} \mathbb{1}_i^{\text{best}} \nabla_{\alpha_i} (1 - \alpha_i)^2 + \frac{1}{2} (1 - \mathbb{1}_i^{\text{best}}) \nabla_{\alpha_i} \alpha_i^2.
\end{aligned} \tag{14}$$

Suppose a batch of examples with a number of $n$ is applied, and the true probability of the primitive $i$ to be the best primitive is $p_i$. The batch accumulated gradient will be

$$\overline{\text{grad}} = \frac{1}{n} \sum_{l=1}^{n} \text{grad}_i^l = \frac{n_t}{2n} \nabla_{\alpha_i} (1 - \alpha_i)^2 \nabla_{\theta_i} \alpha_i + \frac{(n - n_t)}{2n} \nabla_{\alpha_i} \alpha_i^2 \nabla_{\theta_i} \alpha_i \overset{n \to \infty}{=} (\alpha_i - p_i) \nabla_{\theta_i} \alpha_i, \tag{15}$$

where the last formula indicates that $n_t = np_i$ when $n \to \infty$, since the true probability can be approximated by $\frac{n_t}{n}$ in the limit case. Also, as $\nabla_{\theta_i} \alpha_i$ is not always equal to 0, the gradient equals to 0 if and only if $\alpha_i = p_i$. Optimising with the above Eq. (15) is the same as minimising the distance between $\alpha_i$ and $p_i$, with the optimal situation as $\alpha_i = p_i$ when letting the last formula of Eq. (15) be zero. Based the above analyses, we therefore can build a frequency loss in this paper.

The derivation is completed. ∎

### A.3 The Proof of Proposition 2

We employ the *mixture of Gaussian* for the *dynamic* model $q_\theta(\mathbf{z}_c|\mathbf{z}_m)$, which is more flexible than a unimodal and can diversify the output. More formally, we have

$$q_\theta(\mathbf{z}_c|\mathbf{z}_m) = \sum_{i=1}^{k} \alpha_i \mathcal{N}(\mathbf{z}_c|\boldsymbol{\mu}_i^{\hat{\mathbf{z}}_c}, \boldsymbol{\Sigma}_i^{\hat{\mathbf{z}}_c}), \tag{16}$$

where $k$ denotes the number of primitives, and $\alpha_i$ denotes the weight of the $i$-th primitive. Then the KL divergence between $q_\theta(\mathbf{z}_c|\mathbf{z}_m)$ and $p_\phi(\mathbf{z}_c|\mathbf{I}_c)$ becomes

$$\begin{aligned}
&\text{KL}\left[q_\theta(\mathbf{z}_c|\mathbf{z}_m) \| p_\phi(\mathbf{z}_c|\mathbf{I}_c)\right] \\
&= \int_{\mathbf{z}_c} q_\theta(\mathbf{z}_c|\mathbf{z}_m) \log \frac{q_\theta(\mathbf{z}_c|\mathbf{z}_m)}{p_\phi(\mathbf{z}_c|\mathbf{I}_c)} d\mathbf{z}_c \\
&= \int_{\mathbf{z}_c} \sum_{i=1}^{k} \alpha_i \mathcal{N}(\mathbf{z}_c|\boldsymbol{\mu}_i^{\hat{\mathbf{z}}_c}, \boldsymbol{\Sigma}_i^{\hat{\mathbf{z}}_c}) \log \frac{\sum_{i=1}^{k} \alpha_i \mathcal{N}(\mathbf{z}_c|\boldsymbol{\mu}_i^{\hat{\mathbf{z}}_c}, \boldsymbol{\Sigma}_i^{\hat{\mathbf{z}}_c})}{\mathcal{N}(\mathbf{z}_c|\boldsymbol{\mu}^{\mathbf{z}_c}, \boldsymbol{\Sigma}^{\mathbf{z}_c})} d\mathbf{z}_c.
\end{aligned} \tag{17}$$

We further approximate Eq. (17) by the back-propogate-max-operation with the frequency loss. We will have

$$\text{KL}\left[q_\theta(\mathbf{z}_c|\mathbf{z}_m) \| p_\phi(\mathbf{z}_c|\mathbf{I}_c)\right] \approx \int_{\mathbf{z}_c} \mathcal{N}(\mathbf{z}_c|\boldsymbol{\mu}_j^{\hat{\mathbf{z}}_c}, \boldsymbol{\Sigma}_j^{\hat{\mathbf{z}}_c}) \log \frac{\mathcal{N}(\mathbf{z}_c|\boldsymbol{\mu}_j^{\hat{\mathbf{z}}_c}, \boldsymbol{\Sigma}_j^{\hat{\mathbf{z}}_c})}{\mathcal{N}(\mathbf{z}_c|\boldsymbol{\mu}^{\mathbf{z}_c}, \boldsymbol{\Sigma}^{\mathbf{z}_c})} d\mathbf{z}_c, \tag{18}$$

where $j = \operatorname{argmin}_i \operatorname{KL}\left[\mathcal{N}(\mathbf{z}_c|\boldsymbol{\mu}_i^{\hat{\mathbf{z}}_c}, \boldsymbol{\Sigma}_i^{\hat{\mathbf{z}}_c}) \| \mathcal{N}(\mathbf{z}_c|\boldsymbol{\mu}^{\mathbf{z}_c}, \boldsymbol{\Sigma}^{\mathbf{z}_c})\right]$. We then have

$$
\int_{\mathbf{z}_c} \mathcal{N}(\mathbf{z}_c|\boldsymbol{\mu}_j^{\hat{\mathbf{z}}_c}, \boldsymbol{\Sigma}_j^{\hat{\mathbf{z}}_c}) \log \frac{\mathcal{N}(\mathbf{z}_c|\boldsymbol{\mu}_j^{\hat{\mathbf{z}}_c}, \boldsymbol{\Sigma}_j^{\hat{\mathbf{z}}_c})}{\mathcal{N}(\mathbf{z}_c|\boldsymbol{\mu}^{\mathbf{z}_c}, \boldsymbol{\Sigma}^{\mathbf{z}_c})} \mathrm{d}\mathbf{z}_c
$$

$$
= \int_{\mathbf{z}_c} \mathcal{N}(\mathbf{z}_c|\boldsymbol{\mu}_j^{\hat{\mathbf{z}}_c}, \boldsymbol{\Sigma}_j^{\hat{\mathbf{z}}_c}) \log \frac{(2\pi)^{-\frac{d}{2}}|\boldsymbol{\Sigma}_j^{\hat{\mathbf{z}}_c}|^{-\frac{1}{2}} \exp\left[-\frac{1}{2}(\mathbf{z}_c - \boldsymbol{\mu}_j^{\hat{\mathbf{z}}_c})^\top (\boldsymbol{\Sigma}_j^{\hat{\mathbf{z}}_c})^{-1}(\mathbf{z}_c - \boldsymbol{\mu}_j^{\hat{\mathbf{z}}_c})\right]}{(2\pi)^{-\frac{d}{2}}|\boldsymbol{\Sigma}^{\mathbf{z}_c}|^{-\frac{1}{2}} \exp\left[-\frac{1}{2}(\mathbf{z}_c - \boldsymbol{\mu}^{\mathbf{z}_c})^\top (\boldsymbol{\Sigma}^{\mathbf{z}_c})^{-1}(\mathbf{z}_c - \boldsymbol{\mu}^{\mathbf{z}_c})\right]} \mathrm{d}\mathbf{z}_c
$$

$$
= \int_{\mathbf{z}_c} \mathcal{N}(\mathbf{z}_c|\boldsymbol{\mu}_j^{\hat{\mathbf{z}}_c}, \boldsymbol{\Sigma}_j^{\hat{\mathbf{z}}_c}) \log \frac{|\boldsymbol{\Sigma}_j^{\hat{\mathbf{z}}_c}|^{-\frac{1}{2}} \exp\left[-\frac{1}{2}(\mathbf{z}_c - \boldsymbol{\mu}_j^{\hat{\mathbf{z}}_c})^\top (\boldsymbol{\Sigma}_j^{\hat{\mathbf{z}}_c})^{-1}(\mathbf{z}_c - \boldsymbol{\mu}_j^{\hat{\mathbf{z}}_c})\right]}{|\boldsymbol{\Sigma}^{\mathbf{z}_c}|^{-\frac{1}{2}} \exp\left[-\frac{1}{2}(\mathbf{z}_c - \boldsymbol{\mu}^{\mathbf{z}_c})^\top (\boldsymbol{\Sigma}^{\mathbf{z}_c})^{-1}(\mathbf{z}_c - \boldsymbol{\mu}^{\mathbf{z}_c})\right]} \mathrm{d}\mathbf{z}_c
$$

$$
= \int_{\mathbf{z}_c} \mathcal{N}(\mathbf{z}_c|\boldsymbol{\mu}_j^{\hat{\mathbf{z}}_c}, \boldsymbol{\Sigma}_j^{\hat{\mathbf{z}}_c})\{-\frac{1}{2}\log\frac{|\boldsymbol{\Sigma}_j^{\hat{\mathbf{z}}_c}|}{|\boldsymbol{\Sigma}^{\mathbf{z}_c}|} + \frac{1}{2}[(\mathbf{z}_c - \boldsymbol{\mu}^{\mathbf{z}_c})^\top (\boldsymbol{\Sigma}^{\mathbf{z}_c})^{-1}(\mathbf{z}_c - \boldsymbol{\mu}^{\mathbf{z}_c}) - (\mathbf{z}_c - \boldsymbol{\mu}_j^{\hat{\mathbf{z}}_c})^\top (\boldsymbol{\Sigma}_j^{\hat{\mathbf{z}}_c})^{-1}(\mathbf{z}_c - \boldsymbol{\mu}_j^{\hat{\mathbf{z}}_c})]\}\mathrm{d}\mathbf{z}_c
$$

$$
= -\frac{1}{2}\log\frac{|\boldsymbol{\Sigma}_j^{\hat{\mathbf{z}}_c}|}{|\boldsymbol{\Sigma}^{\mathbf{z}_c}|} + \frac{1}{2}\int_{\mathbf{z}_c} \mathcal{N}(\mathbf{z}_c|\boldsymbol{\mu}_j^{\hat{\mathbf{z}}_c}, \boldsymbol{\Sigma}_j^{\hat{\mathbf{z}}_c})\Big[\mathbf{z}_c^\top (\boldsymbol{\Sigma}^{\mathbf{z}_c})^{-1}\mathbf{z}_c - \mathbf{z}_c^\top (\boldsymbol{\Sigma}^{\mathbf{z}_c})^{-1}\boldsymbol{\mu}^{\mathbf{z}_c} - \boldsymbol{\mu}^{\mathbf{z}_c\top}(\boldsymbol{\Sigma}^{\mathbf{z}_c})^{-1}\mathbf{z}_c + \boldsymbol{\mu}^{\mathbf{z}_c\top}(\boldsymbol{\Sigma}^{\mathbf{z}_c})^{-1}\boldsymbol{\mu}^{\mathbf{z}_c}
$$

$$
- (\mathbf{z}_c^\top (\boldsymbol{\Sigma}_j^{\hat{\mathbf{z}}_c})^{-1}\mathbf{z}_c - \mathbf{z}_c^\top (\boldsymbol{\Sigma}_j^{\hat{\mathbf{z}}_c})^{-1}\boldsymbol{\mu}_j^{\hat{\mathbf{z}}_c} - \boldsymbol{\mu}_j^{\hat{\mathbf{z}}_c\top}(\boldsymbol{\Sigma}_j^{\hat{\mathbf{z}}_c})^{-1}\mathbf{z}_c + \boldsymbol{\mu}_j^{\hat{\mathbf{z}}_c\top}(\boldsymbol{\Sigma}_j^{\hat{\mathbf{z}}_c})^{-1}\boldsymbol{\mu}_j^{\hat{\mathbf{z}}_c})\Big]\mathrm{d}\mathbf{z}_c.
$$

$$(19)$$

Suppose the covariance is a diagonal matrix [21], we have

$$
\int_{\mathbf{z}_c} \mathcal{N}(\mathbf{z}_c|\boldsymbol{\mu}_j^{\hat{\mathbf{z}}_c}, \boldsymbol{\Sigma}_j^{\hat{\mathbf{z}}_c})\mathbf{z}_c^\top \mathbf{z}_c \mathrm{d}\mathbf{z}_c = \operatorname{tr}(\boldsymbol{\Sigma}_j^{\hat{\mathbf{z}}_c}) + \boldsymbol{\mu}_j^{\hat{\mathbf{z}}_c\top}\boldsymbol{\mu}_j^{\hat{\mathbf{z}}_c}, \tag{20}
$$

where $\operatorname{tr}(\cdot)$ denotes the trace of a matrix. Then, Eq. (19) would be:

$$
\int_{\mathbf{z}_c} \mathcal{N}(\mathbf{z}_c|\boldsymbol{\mu}_j^{\hat{\mathbf{z}}_c}, \boldsymbol{\Sigma}_j^{\hat{\mathbf{z}}_c}) \log \frac{\mathcal{N}(\mathbf{z}_c|\boldsymbol{\mu}_j^{\hat{\mathbf{z}}_c}, \boldsymbol{\Sigma}_j^{\hat{\mathbf{z}}_c})}{\mathcal{N}(\mathbf{z}_c|\boldsymbol{\mu}^{\mathbf{z}_c}, \boldsymbol{\Sigma}^{\mathbf{z}_c})} \mathrm{d}\mathbf{z}_c
$$

$$
= -\frac{1}{2}\log\frac{|\boldsymbol{\Sigma}_j^{\hat{\mathbf{z}}_c}|}{|\boldsymbol{\Sigma}^{\mathbf{z}_c}|} - \frac{1}{2}\Big[\operatorname{tr}\left((\boldsymbol{\Sigma}^{\mathbf{z}_c})^{-1}\boldsymbol{\Sigma}_j^{\hat{\mathbf{z}}_c}\right) + \boldsymbol{\mu}_j^{\hat{\mathbf{z}}_c\top}(\boldsymbol{\Sigma}^{\mathbf{z}_c})^{-1}\boldsymbol{\mu}_j^{\hat{\mathbf{z}}_c} - \boldsymbol{\mu}_j^{\hat{\mathbf{z}}_c\top}(\boldsymbol{\Sigma}^{\mathbf{z}_c})^{-1}\boldsymbol{\mu}^{\mathbf{z}_c} - \boldsymbol{\mu}^{\mathbf{z}_c\top}(\boldsymbol{\Sigma}^{\mathbf{z}_c})^{-1}\boldsymbol{\mu}_j^{\hat{\mathbf{z}}_c}
$$

$$
+ \boldsymbol{\mu}^{\mathbf{z}_c\top}(\boldsymbol{\Sigma}^{\mathbf{z}_c})^{-1}\boldsymbol{\mu}^{\mathbf{z}_c} + \operatorname{tr}\left((\boldsymbol{\Sigma}_j^{\hat{\mathbf{z}}_c})^{-1}\boldsymbol{\Sigma}_j^{\hat{\mathbf{z}}_c}\right) + \boldsymbol{\mu}_j^{\hat{\mathbf{z}}_c\top}(\boldsymbol{\Sigma}_j^{\hat{\mathbf{z}}_c})^{-1}\boldsymbol{\mu}_j^{\hat{\mathbf{z}}_c} - \boldsymbol{\mu}_j^{\hat{\mathbf{z}}_c\top}(\boldsymbol{\Sigma}_j^{\hat{\mathbf{z}}_c})^{-1}\boldsymbol{\mu}_j^{\hat{\mathbf{z}}_c} - \boldsymbol{\mu}_j^{\hat{\mathbf{z}}_c\top}(\boldsymbol{\Sigma}_j^{\hat{\mathbf{z}}_c})^{-1}\boldsymbol{\mu}_j^{\hat{\mathbf{z}}_c}
$$

$$
+ \boldsymbol{\mu}_j^{\hat{\mathbf{z}}_c\top}(\boldsymbol{\Sigma}_j^{\hat{\mathbf{z}}_c})^{-1}\boldsymbol{\mu}_j^{\hat{\mathbf{z}}_c}\Big]
$$

$$
= -\frac{1}{2}\log\frac{|\boldsymbol{\Sigma}_j^{\hat{\mathbf{z}}_c}|}{|\boldsymbol{\Sigma}^{\mathbf{z}_c}|} + \frac{1}{2}\operatorname{tr}\left((\boldsymbol{\Sigma}^{\mathbf{z}_c})^{-1}\boldsymbol{\Sigma}_j^{\hat{\mathbf{z}}_c}\right) - \frac{1}{2}(\boldsymbol{\mu}_j^{\hat{\mathbf{z}}_c} - \boldsymbol{\mu}^{\mathbf{z}_c})^\top (\boldsymbol{\Sigma}^{\mathbf{z}_c})^{-1}(\boldsymbol{\mu}_j^{\hat{\mathbf{z}}_c} - \boldsymbol{\mu}^{\mathbf{z}_c}).
$$

$$(21)$$

The proof is completed. ∎

### A.4  The Fully Optimization Expression of Proposition 1

As discussed in the main paper, we have

$$
L_{\mathrm{P}} = \operatorname{KL}\left[q_\psi(\mathbf{I}_o, \mathbf{z}_m, \mathbf{z}_c|\mathbf{I}_m) \| p_\phi(\mathbf{I}_o, \mathbf{z}_m, \mathbf{z}_c|\mathbf{I}_m, \mathbf{I}_c)\right]
$$

$$
= \underbrace{\mathbb{E}_{(\mathbf{z}_m, \mathbf{z}_c)\sim q_\psi(\mathbf{z}_m, \mathbf{z}_c|\mathbf{I}_m)}\operatorname{KL}\left[q_\psi(\mathbf{I}_o|\mathbf{z}_m, \mathbf{z}_c) \| p_\phi(\mathbf{I}_o|\mathbf{I}_m, \mathbf{I}_c)\right]}_{\text{\textcircled{a}}}
$$

$$
+ \underbrace{\mathbb{E}_{\mathbf{z}_m\sim q_\psi(\mathbf{z}_m|\mathbf{I}_m)}\operatorname{KL}\left[q_\theta(\mathbf{z}_c|\mathbf{z}_m) \| p_\phi(\mathbf{z}_c|\mathbf{I}_c)\right]}_{\text{\textcircled{b}}} \tag{22}
$$

$$
+ \underbrace{\operatorname{KL}\left[q_\psi(\mathbf{z}_m|\mathbf{I}_m) \| p_\phi(\mathbf{z}_m|\mathbf{I}_m)\right]}_{\text{\textcircled{c}}}.
$$

As Eq. (22) shows, the item $\textcircled{c}$ may be hard to implement, because we cannot know the real $p_\phi(\mathbf{z}_m|\mathbf{I}_m)$, which is used to restrain $q_\psi(\mathbf{z}_m|\mathbf{I}_m)$ outputted by a neural network.

However, inspired by [21], we can employ a variational evidence lower bound (ELBO) to implement the item ©. Firstly, we assume that there exists a ground truth model $p(\mathbf{I}_m)$, which is a constant given $\mathbf{I}_m$ and does not depend on $\mathbf{z}_m$. Then, we can derive the relationship between $p(\mathbf{I}_m)$ and ©:

$$
\begin{aligned}
\log p(\mathbf{I}_m) &= \mathbb{E}_{\mathbf{z}_m \sim q_\psi(\mathbf{z}_m|\mathbf{I}_m)}[\log p(\mathbf{I}_m)] \\
&= \mathbb{E}_{\mathbf{z}_m \sim q_\psi(\mathbf{z}_m|\mathbf{I}_m)}[\log \frac{p(\mathbf{I}_m|\mathbf{z}_m)p(\mathbf{z}_m)}{p_\phi(\mathbf{z}_m|\mathbf{I}_m)}] \\
&= \mathbb{E}_{\mathbf{z}_m \sim q_\psi(\mathbf{z}_m|\mathbf{I}_m)}[\log \frac{p(\mathbf{I}_m|\mathbf{z}_m)p(\mathbf{z}_m)}{p_\phi(\mathbf{z}_m|\mathbf{I}_m)} \frac{q_\psi(\mathbf{z}_m|\mathbf{I}_m)}{q_\psi(\mathbf{z}_m|\mathbf{I}_m)}] \\
&= \mathbb{E}_{\mathbf{z}_m}[\log p(\mathbf{I}_m|\mathbf{z}_m)] - \mathbb{E}_{\mathbf{z}_m}[\log \frac{q_\psi(\mathbf{z}_m|\mathbf{I}_m)}{p(\mathbf{z}_m)}] + \mathbb{E}_{\mathbf{z}_m}[\log \frac{q_\psi(\mathbf{z}_m|\mathbf{I}_m)}{p_\phi(\mathbf{z}_m|\mathbf{I}_m)}] \\
&= \mathbb{E}_{\mathbf{z}_m}[\log p(\mathbf{I}_m|\mathbf{z}_m)] - \mathrm{KL}[q_\psi(\mathbf{z}_m|\mathbf{I}_m)\|p(\mathbf{z}_m)] + \mathrm{KL}[q_\psi(\mathbf{z}_m|\mathbf{I}_m)\|p_\phi(\mathbf{z}_m|\mathbf{I}_m)] \\
&= \underbrace{\mathbb{E}_{\mathbf{z}_m}[\log p(\mathbf{I}_m|\mathbf{z}_m)] - \mathrm{KL}[q_\psi(\mathbf{z}_m|\mathbf{I}_m)\|p(\mathbf{z}_m)]}_{\text{ELBO}} + ©
\end{aligned}
\tag{23}
$$

Thus, © is equal to $\log p(\mathbf{I}_m) - \text{ELBO}$. Since $\log p(\mathbf{I}_m)$ is a constant, minimizing © is equal to maximize the ELBO.

Accordingly, the fully optimization expression of Theorem 1 is below:

$$
\begin{aligned}
L_{\mathrm{P}} &= \mathrm{KL}\left[q_\psi(\mathbf{I}_o, \mathbf{z}_m, \mathbf{z}_c|\mathbf{I}_m)\|p_\phi(\mathbf{I}_o, \mathbf{z}_m, \mathbf{z}_c|\mathbf{I}_m, \mathbf{I}_c)\right] \\
&= \underbrace{\mathbb{E}_{(\mathbf{z}_m, \mathbf{z}_c) \sim q_\psi(\mathbf{z}_m, \mathbf{z}_c|\mathbf{I}_m)} \mathrm{KL}\left[q_\psi(\mathbf{I}_o|\mathbf{z}_m, \mathbf{z}_c)\|p_\phi(\mathbf{I}_o|\mathbf{I}_m, \mathbf{I}_c)\right]}_{\text{ⓐ}} \\
&\quad + \underbrace{\mathbb{E}_{\mathbf{z}_m \sim q_\psi(\mathbf{z}_m|\mathbf{I}_m)} \mathrm{KL}\left[q_\theta(\mathbf{z}_c|\mathbf{z}_m)\|p_\phi(\mathbf{z}_c|\mathbf{I}_c)\right]}_{\text{ⓑ}} \\
&\quad - \underbrace{\mathbb{E}_{\mathbf{z}_m \sim q_\psi(\mathbf{z}_m|\mathbf{I}_m)}[\log p(\mathbf{I}_m|\mathbf{z}_m)] + \mathrm{KL}[q_\psi(\mathbf{z}_m|\mathbf{I}_m)\|p(\mathbf{z}_m)]}_{\text{ⓒ}},
\end{aligned}
\tag{24}
$$

∎

# B  Advance of PICMM Over PIC

The illustration of our method is shown in Figure 11.

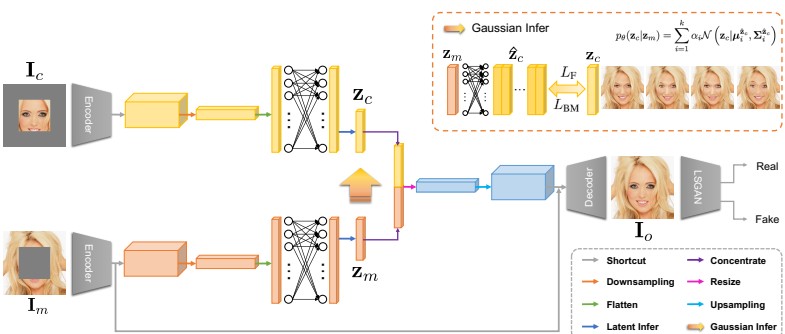

Figure 11: The algorithm framework of the proposed method.

We also summarize the differences between PIC and our method as follows:

(1) Our method has *better interpretability* in pluralistic image completion than PIC, which is highlighted in this paper.

(2) GMM is used in our method, which better promotes diversity than PIC. More importantly, the inherent parameters for diversity are *task-related*, rather than *task-agnostic* in PIC.

(3) Our method uses KL divergence for the whole objective. We decompose the divergence and then minimize different decomposition terms, which supports our theory. While, PIC only uses KL divergence as a single loss in its objective.

(4) Our method exploits the same CNN backbone as PIC. In all quantitative comparisons, our method outperforms PIC.

# C  Supplementary Experimental Results on Diversity Analyses

In the main paper, we state that our method is not limited to only generate $k$ image completion results. Instead, we can sample from the $k$ primitives of GMM to generate a different number of images. The results are provided in Figure 12 and 13. The images in the $i$-th row of Figure 12 and 13 are obtained by sampling from the $i$-th primitive of GMM. As can be seen, the images in the same rows are less diverse than the images in the different rows, since one primitive has a limited capacity. Multiple primitives can better meet the diversity needs. We are able to sample from different primitives to achieve the better diversity in pluralistic image completion.

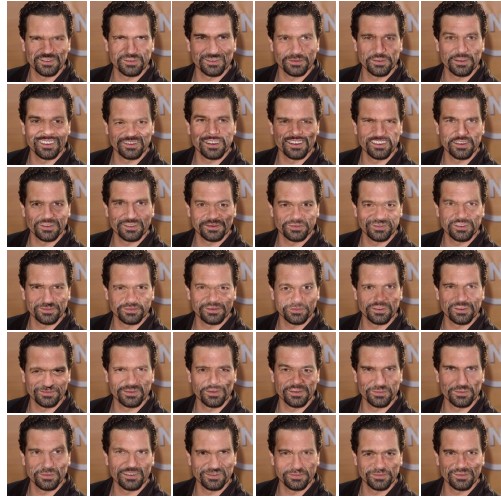

Figure 12: Pluralistic image completion results of our method by sampling from $k$ primitives ($k = 6$). With each primitive, six diverse images are generated.

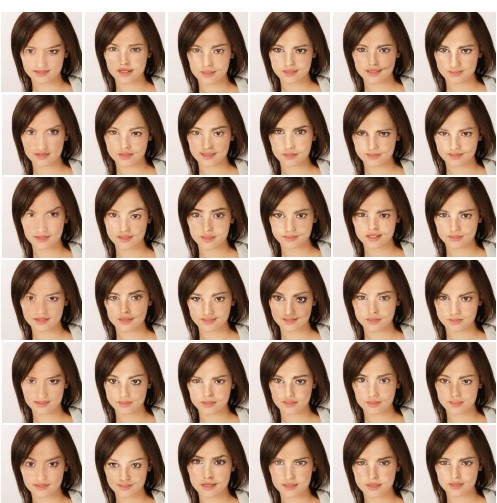

Figure 13: Pluralistic image completion results of our method by sampling from $k$ primitives ($k = 6$). With each primitive, six diverse images are generated.

# D  Supplementary Pluralistic Completion Result Comparisons

In the main paper, we provide some comparisons of pluralistic completion results with state-of-the-art methods. We provide more comparisons here. The comparison methods include DFv2[3], EC[4], MED[5], PIC[6], and ICT[7]. The experimental results on CelebA-HQ are provided in Figures 14 and 15. The experimental results on FFHQ, Paris StreetView, and Places2 are shown in Figures 16, 17, and 18 respectively. In addition, the experimental results on ImageNet are presented in Figure 19.

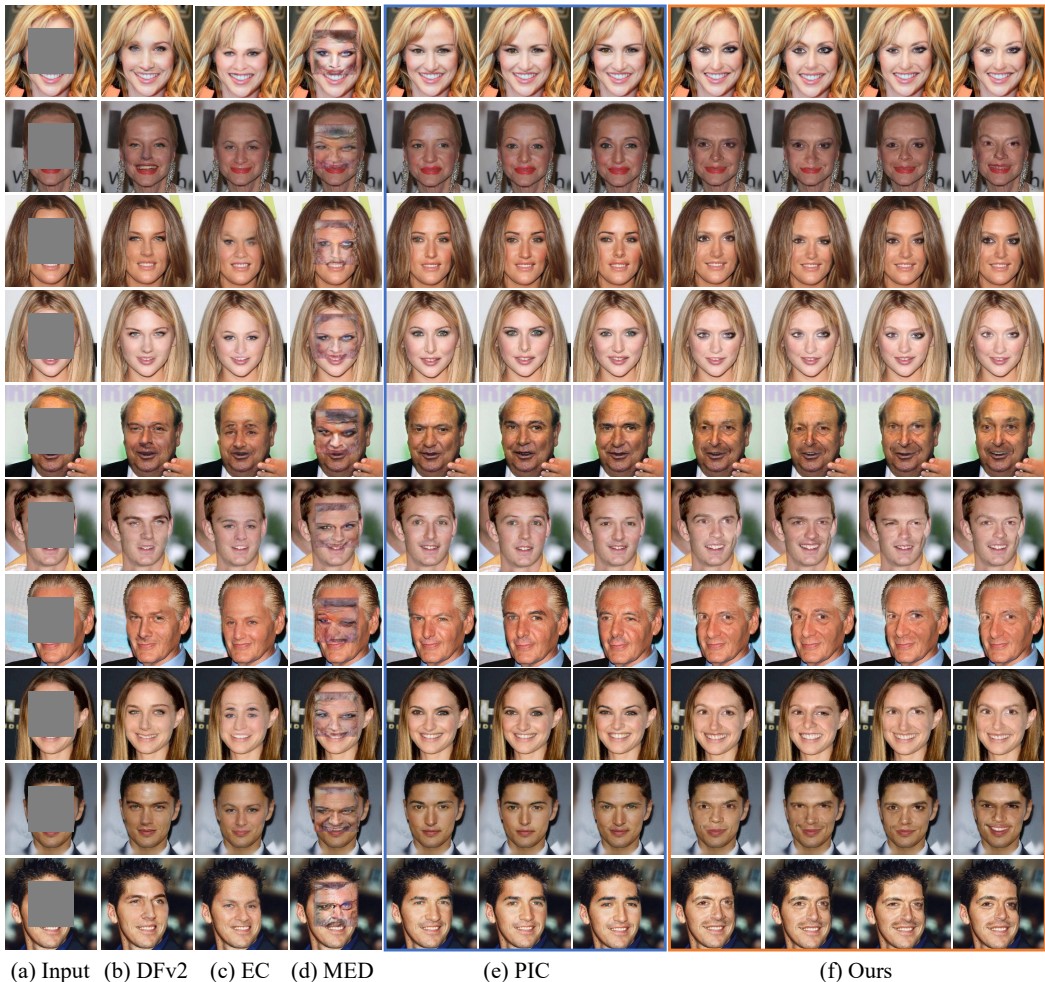

(a) Input  (b) DFv2  (c) EC  (d) MED  (e) PIC  (f) Ours

Figure 14: Pluralistic image completion results comparison with baselines. The original images come from CelebA-HQ. Best viewed by zooming in.

[3]https://github.com/JiahuiYu/generative_inpainting
[4]https://github.com/knazeri/edge-connect
[5]https://github.com/KumapowerLIU/Rethinking-Inpainting-MEDFE
[6]https://github.com/lyndonzheng/Pluralistic-Inpainting
[7]https://github.com/raywzy/ICT

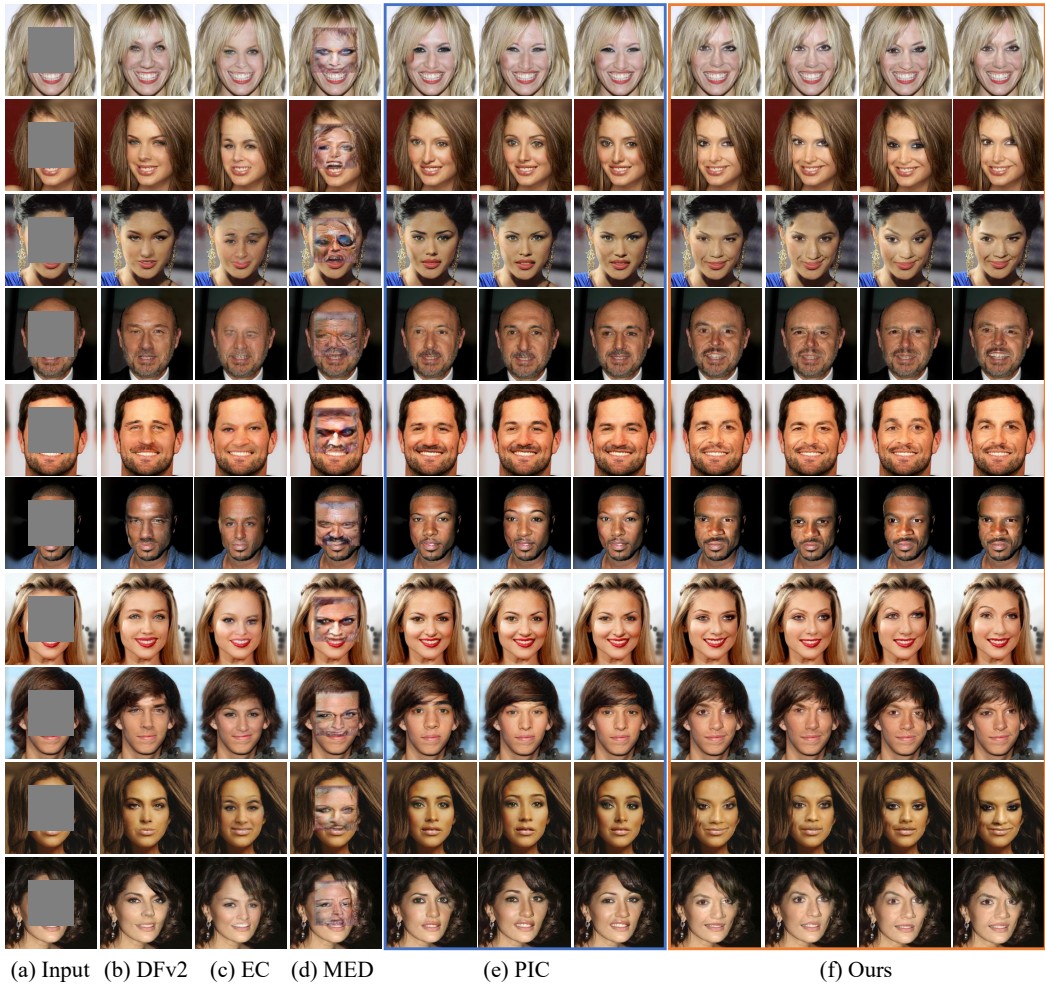

(a) Input  (b) DFv2  (c) EC  (d) MED  (e) PIC  (f) Ours

Figure 15: Pluralistic image completion results comparison with baselines. The original images come from CelebA-HQ. Best viewed by zooming in.

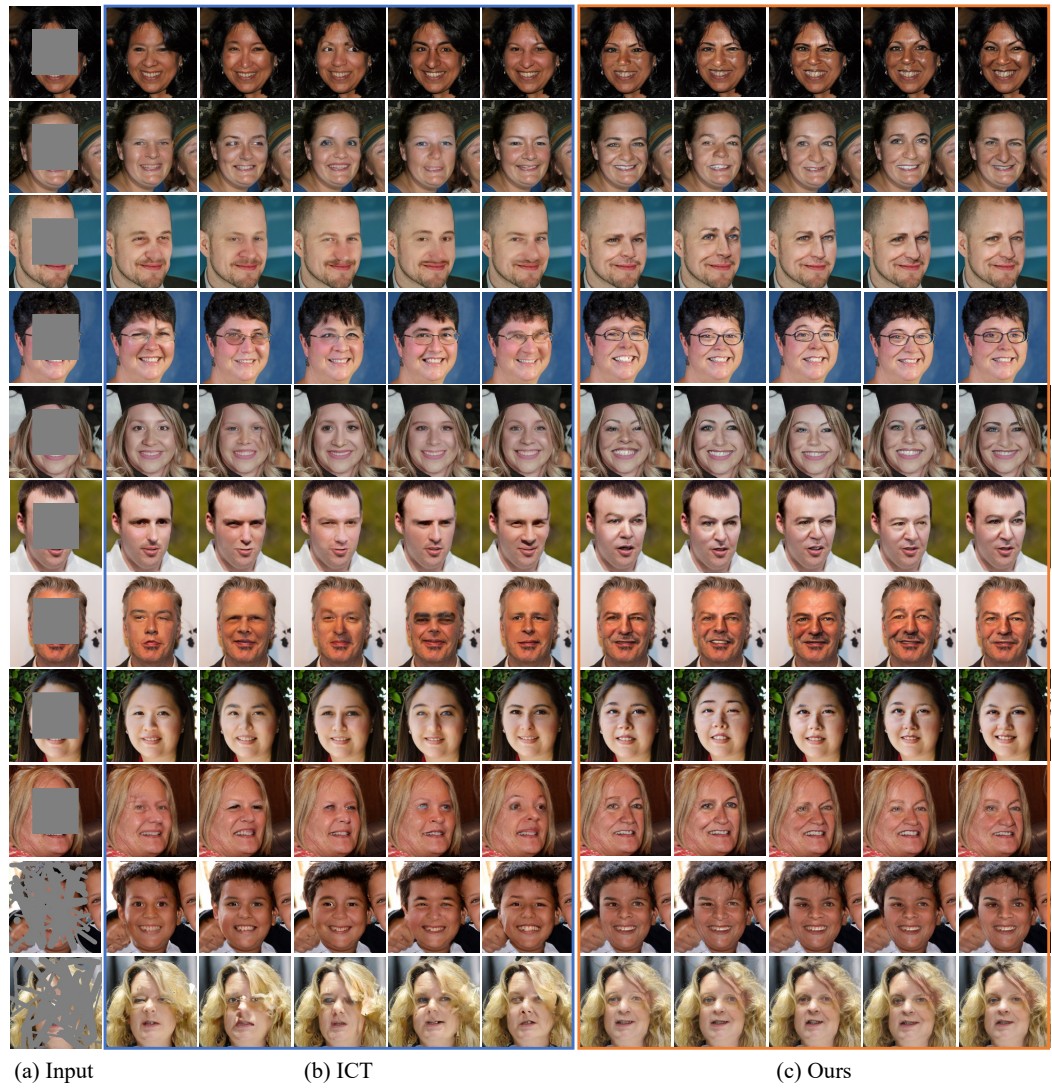

(a) Input       (b) ICT       (c) Ours

Figure 16: Pluralistic image completion results comparison with baselines. The original images come from FFHQ. Best viewed by zooming in.

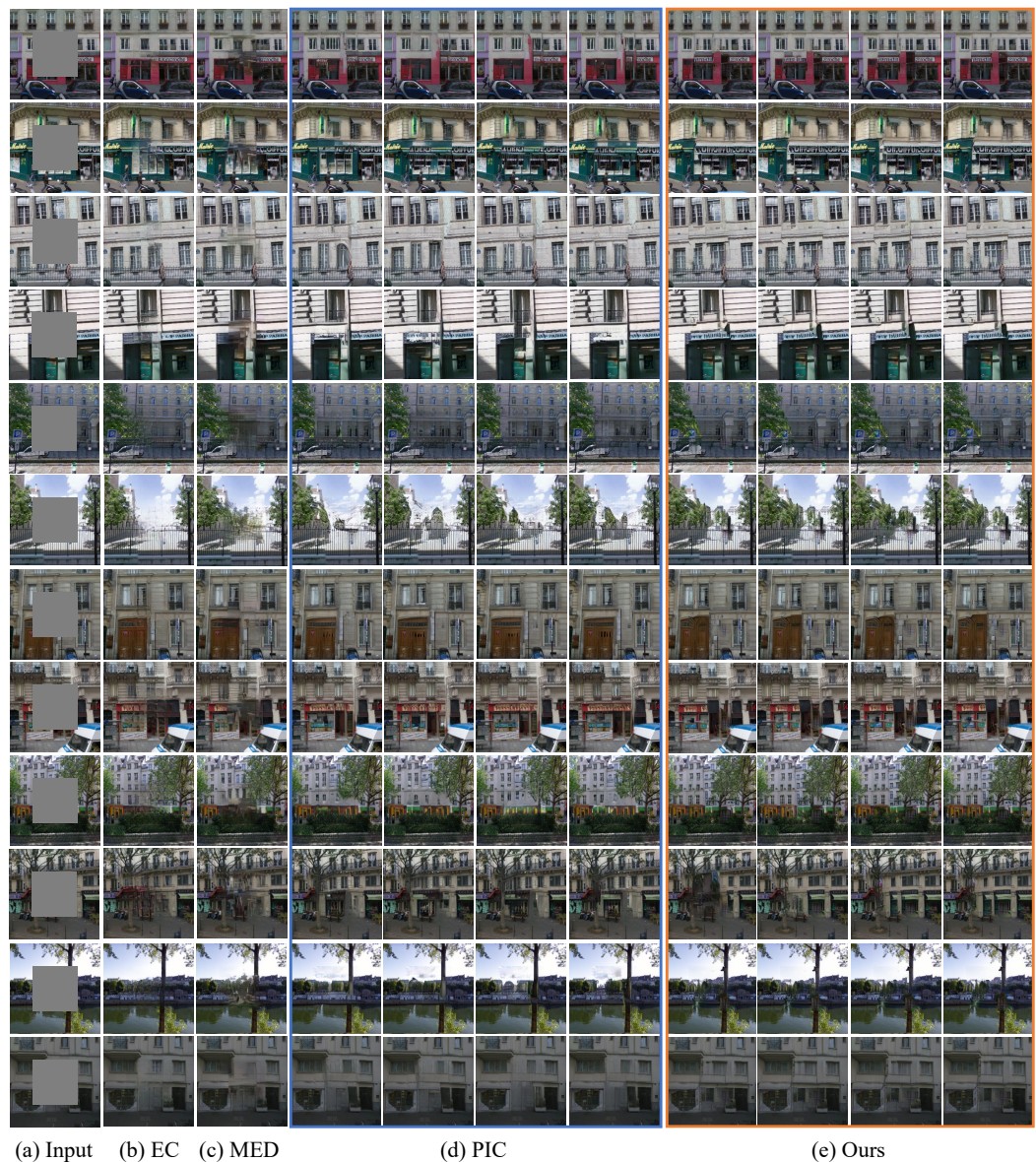

(a) Input    (b) EC   (c) MED        (d) PIC              (e) Ours

Figure 17: Pluralistic image completion results comparison with baselines. The original images come from Paris StreetView. Best viewed by zooming in.

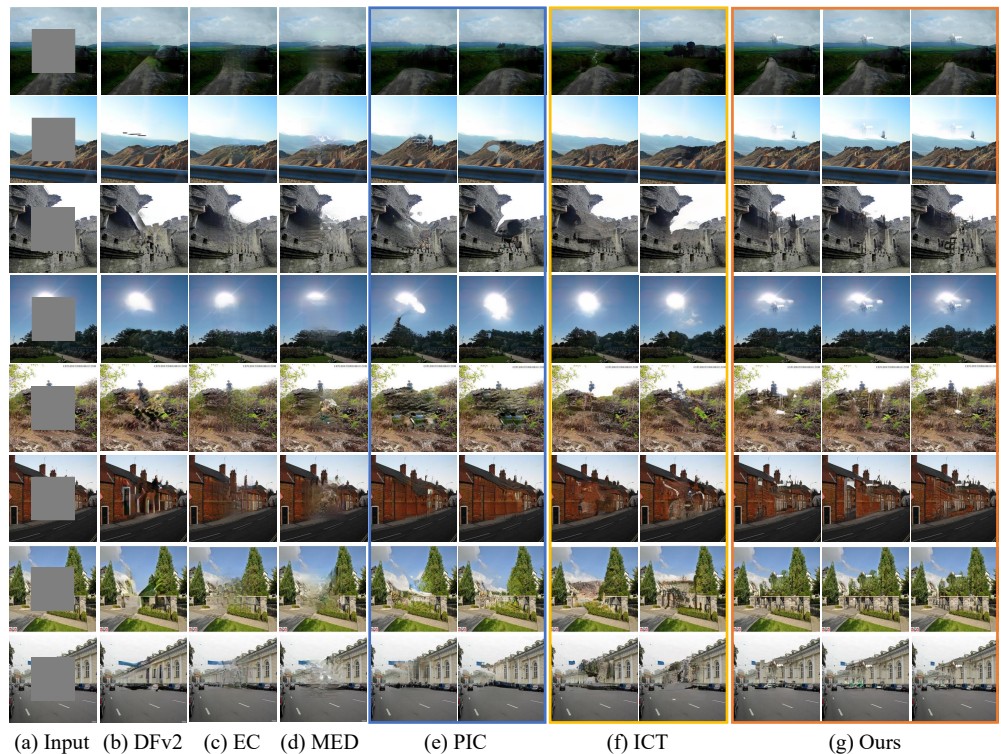

(a) Input    (b) DFv2    (c) EC    (d) MED    (e) PIC    (f) ICT    (g) Ours

Figure 18: Pluralistic image completion results comparison with baselines. The original images come from Places2. Best viewed by zooming in.

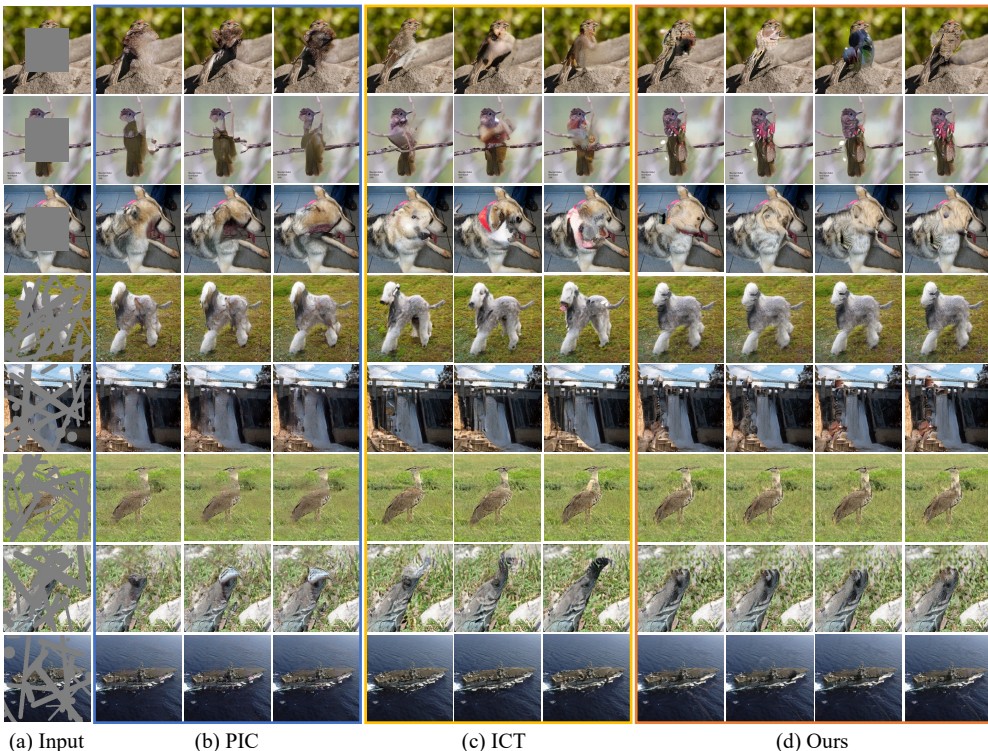

(a) Input    (b) PIC    (c) ICT    (d) Ours

Figure 19: Pluralistic image completion results comparison with baselines. The original images come from ImageNet. Best viewed by zooming in.

# E Ablation Study

To help understand the contribution of the different components in the proposed method, we provide ablation studies here. Specifically, experiments are conducted on the dataset CelebA-HQ with the setting of center masking. The number of primitives is set to $k = 6$. It should be noted that, without the loss $L_R$, the image completion is unsupervised, resulting in rather unrealistic completed images. The loss $L_R$ is hence added by default. The results of ablation studies are provided in Table 5, which demonstrate the effectiveness of different components.

| $L_F$ | $L_{BM}$ | $L_{ELBO}$ | $L_A$ | PSNR ↑ | SSIM ↑ | MAE ↓ | FID ↓ | LPIPS ↑ | DIV-FID ↑ |
|---|---|---|---|---|---|---|---|---|---|
| ✓ | ✓ | ✓ | ✓ | 27.098 | 0.964 | 0.0196 | 11.021 | 0.098 | 27.923 |
| ✓ | ✓ | ✓ | - | 27.034 | 0.961 | 0.0194 | 11.193 | 0.095 | 27.534 |
| ✓ | ✓ | - | - | 27.035 | 0.961 | 0.0198 | 11.202 | 0.094 | 27.068 |
| ✓ | - | - | - | 26.251 | 0.922 | 0.0232 | 11.589 | 0.086 | 26.238 |

Table 5: Ablation study of PICMM.

# F Results of Comparison with More Image Completion Methods

To make the experimental comparison more comprehensive and more convincing, we add the baseline VQ-VAE [37] and AOT-GAN [54]. The results of comparing our PICMM with the two baselines are shown in Table 6, which verify the effectiveness of our method clearly.

| Method | PSNR ↑ | SSIM ↑ | MAE ↓ | FID ↓ | LPIPS ↑ | DIV-FID ↑ |
|---|---|---|---|---|---|---|
| VQ-VAE [37] | 24.031 | 0.819 | 0.0309 | 30.747 | 0.083 | 23.676 |
| AOT-GAN [54] | 25.455 | 0.831 | 0.0233 | 26.427 | 0.089 | 25.843 |
| PICMM$^\dagger$ | **25.467** | **0.838** | **0.0223** | **24.334** | **0.091** | **28.033** |

Table 6: The results of comparing our PICMM with VQ-VAE and AOT-GAN. The experiments are conducted on Places2 with random masking.

# G Evaluations on the Number of Primitives

We supplement the experiments with different number of primitives, *i.e.*, $k = 1, 2, 3, 4, 5$. The experiments are conducted on CelebA-HQ using center masks. The implementation details keep the same as the details in the main paper. We provide experimental results in Table 7. The results show that the increase of $k$ can diversify the completed images, without degenerating the performance with respect to common evaluation metrics.

| $k$ | PSNR ↑ | SSIM ↑ | MAE ↓ | FID ↓ | LPIPS ↑ | DIV-FID ↑ |
|---|---|---|---|---|---|---|
| 1 | 26.792 | 0.961 | 0.0201 | 12.011 | 0.084 | 24.223 |
| 2 | 26.979 | 0.965 | 0.0198 | 11.579 | 0.089 | 26.001 |
| 3 | 27.152 | 0.960 | 0.0197 | 11.362 | 0.092 | 27.314 |
| 4 | 26.943 | 0.964 | 0.0197 | 11.638 | 0.096 | 27.540 |
| 5 | 27.121 | 0.966 | 0.0195 | 11.157 | 0.098 | 27.756 |

Table 7: Evaluations on the number of primitives $k$.

# H  Higher-Resolution Image Completion Results

Before this section, we resize the image to $256 \times 256$ for image completion. Here, to show that our method can generate higher-resolution images, we set the image size to $512 \times 512$. Experiments are performed on the datasets FFHQ. The results are provided in Figure 20.

# I  Possible Negative Impacts and Limitation

For possible negative impacts, since our PICMM is a method based on generative models, like a large number of methods of this kind, generated images with our method may danger to the security of AI systems, with the form of deep fakes. There are some advanced works targeting the issue, *e.g.*, [32, 34], which are helpful to handle the possible negative impacts.

For the limitation, our PICMM does not outperform ICT sometimes with respect to common metrics, since ICT uses a much larger transformer-based structure and iterative Gibbs sampling, which make ICT's inference slow. For the people preferring the performance on common metrics, this would be a limitation of our PICMM.

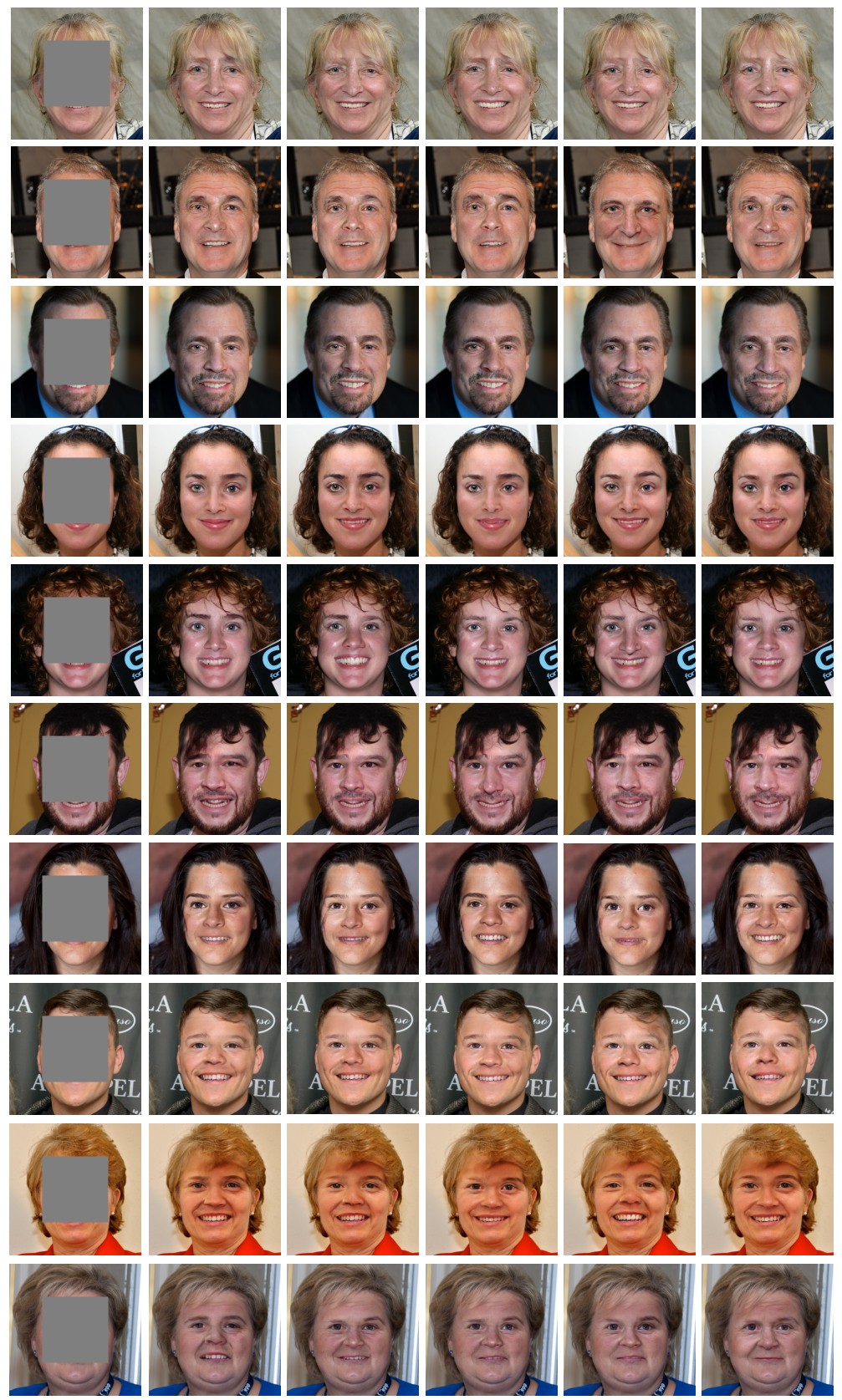

Figure 20: Pluralistic image completion results by our method. The original images come from FFHQ. The image size is $512 \times 512$. Best viewed by zooming in.