# OpenReview forum: "Pluralistic Image Completion with Gaussian Mixture Models"
_NeurIPS.cc/2022/Conference — NeurIPS 2022 Accept_

### Official Review · Reviewer_Nbcm · 2022-07-08

**Rating:** 6
**Confidence:** 4
**Soundness:** 3 good
**Presentation:** 4 excellent
**Contribution:** 3 good

**Summary:**

This paper presents a nice theoretical framework to provide a diverse set of plausible images that come from the reconstruction process under occlusion. The proposed approach aims to deal with diversity by using  a mixture of experts that are based on a mixture of Gaussians. Several comprehensive experiments seem to testify the usefulness of the proposed approach.

**Questions:**

- In section 3.2 it is not so clear how precisely the 1-step Monte Carlo (MC) is done. MC is a way to obtain samples from a distribution in an iterative way. Such an iterative process is not clear. The term “1-step”, refers to what, precisely …
- Also, and related to the above and the MC question, how could it be possible to introduce the Expectation Maximization (EM) algorithm to determine the mixture components of the GMM. Or a sort of …It is well known the EM is able to estimate such components, where the means and the variances of the GMM components are iteratively updated. As it is written in the paper I can figure out the “iterative” update process. Is this done in a single one-step? Is it for this reason that the authors call the algorithm “1-step” ?
- Perhaps, this is the reason why I did not follow exactly the difference \hat{z} and z_c,  … and how these samples are iteratively obtained.
Concerning the above remark, I would recommend to include a figure regarding the architecture of the approach, to better see how the variables play in the framework. I miss the big picture here …
- Determining the number of the mixture components (i.e. “k”) is a model selection problem. Again, taking the EM as a reference procedure, it is known that it is possible to determine the order of the model by using, for instance, the minimum description length. This issue is not addressed in the paper. So the question is how to handle this problem, that is the estimate automatically “k” in an end-to-end methodology as proposed herein.
- Seems that all the components of (2) are modeled by using losses (I’m not sure if this is something related to the above question).  Thus:
-term a) is modeled using L_R + \lambda_A * L_A
-term b) is modeled using L_GMM = LF + L_BM
-term c) is models by using L_ELBO
From the above I did not follow (and did not understand well) the impact of the first term in L_A in eq. (7)

- Another issue that is not clear is written in lines 210-215. By the way, a difficult paragraph to follow.
In the mentioned paragraph it is said that:
- \hat{I^j}_m is generate from  \hat{z^j}_c (line 210)
- \hat{I^j}_m is generated from \hat{I^j}_o (line 211)
So, \hat{I^j}_m has two different sources of generation ? I miss something here
-It would be nice to introduce a “break point” from which the algorithm stars to fail, and to relate this to the GMM …

Minor comments:
I think the acronym “PGM” (line 69) is not introduced previously.


**Limitations:**

See my questions above.

**Strengths And Weaknesses:**

Strengths
This paper contributes to the pluralistic image completion by using a grounded theoretical framework. This is achieved in an elegant way by using a GMM approach as a way to meet diversity. Also, this framework seems to be a generalization of [52], where the latter is focused on only one mixture component of the Normal distribution (unimodal Gaussian) with predefined set parameters
Weaknesses
Although I think this is a valuable contribution to this research field, I have some questions that perhaps constitute the main weaknesses of the present paper.

---

> ### Author Response · Authors · 2022-08-02
> **Response to Reviewer Nbcm**
>
> Thanks for your comments. We address your concerns as follows.
>
> > **Q1 & Q2:** It is not so clear how precisely the 1-step Monte Carlo (MC) is done. The reason why the authors call the algorithm ''1-step'' and related questions.
>
> **A1 & A2:** We use Monte Carlo estimation, which is shown in [1], the word "1-step" means using only one sample from the mixture distribution. Specifically, if we use the Monte Carlo estimates of expectations of some function $f(\mathbf{z})$, such as the posterior distribution $q_{\phi}(\mathbf{z}|\mathbf{x})$, it could be done as $E_{q_{\phi}(\mathbf{z}|\mathbf{x})}[f(\mathbf{z})]\approx\frac{1}{L}\sum_{l=1}^{L}f(\mathbf{z})$. Here, $L$ is the sampling times from the distribution of $q_{\phi}(\mathbf{z}|\mathbf{x})$ and
> it is always set as $L=1$ to enable fast training of models and alleviate the computation burden, see [1], [2], and [3]. Otherwise, sampling more than one time needs to accumulate the gradients for all these samples, which is extremely expensive in computation, and backward the average gradients at last.
>
> For the technical details of the sampling of $\mathbf{z}$, it could be implemented with the reparameterization trick, i.e., $\mathbf{z} = \mu_{\mathbf{z}}(\mathbf{x}) + \sigma_{\mathbf{z}}(\mathbf{x})\epsilon$, where $\epsilon \sim p(\epsilon) = \mathcal{N}(0, \mathbf{I})$, $\mu_{\mathbf{z}}(\mathbf{x})$ and $\sigma_{\mathbf{z}}(\mathbf{x})$ are the parameters of $q(\mathbf{z}|\mathbf{x})=\mathcal{N}(\mu_{\mathbf{z}}(\mathbf{x}), \sigma_{\mathbf{z}}(\mathbf{x}))$. By putting them all together, the 1-step Monte Carlo (MC) is done as follows: $E_{q(\mathbf{z}|\mathbf{x})}[f(\mathbf{z})]=E_{p(\epsilon)}[f(\mu_{\mathbf{z}}(\mathbf{x}) + \sigma_{\mathbf{z}}(\mathbf{x})\epsilon)]\approx\frac{1}{L}\sum_{l=1}^{L}f(\mu_{\mathbf{z}}(\mathbf{x}) + \sigma_{\mathbf{z}}(\mathbf{x})\epsilon)$, where  $\epsilon \sim \mathcal{N}(0, \mathbf{I})$ with $L=1$.
>
> Accordingly, in this paper, there is no direct relationship between the used term ''1-step'' and the determination of the mixture components of the GMM.
>
> > **Q3:** A figure regarding the architecture of the approach is needed.
>
> **A3:** The difference between $\mathbf{z}_c$ and $\hat{\mathbf{z}}_c$  lies in: (1) $\mathbf{z}_c$ is achieved by directly encoding images $\mathbf{I}_c$ into corresponding latent features; (2) $\hat{\mathbf{z}}_c$ is inferred from $\mathbf{z}_m$ (Eq. (3) and Eq. (4)), where one of the primitives of $\mathbf{z}_m$ represents a $\hat{\mathbf{z}}_c$. When the number of the primitives is $k$, we then have $\hat{\mathbf{z}}^{(1)}_c,\hat{\mathbf{z}}^{(2)}_c,\ldots,\hat{\mathbf{z}}^{(k)}_c$.
>
> Thanks for your suggestion. Due to the limited page of the main paper, we provide the algorithm framework of the proposed method in Figure 11 of Appendix B.
>
> > **Q4:** How to handle this problem, that is the estimate automatically $k$ in an end-to-end methodology as proposed herein.
>
> **A4:** If we set $k$ too large, we observed that some of the experts will gradually be inactive, i.e., the Categorical parameters output by the gating function corresponding to these experts, are very small. One possible solution is that we can find these inactive experts and choose the maximum value of active experts for all data points as the value of $k$.
>
> > **Q5:** The impact of the first term in $L_{\text{A}}$ in Eq. (7).
>
> **A5:** The first term in $L_{\text{A}}$ in Eq. (7) can be interpreted as an additional style and reality constraint for the term (a) in Eq. (2), i.e., the likelihood. The philosophy of this constraint can be referred to [4] and [5].
>
> > **Q6:** On the two different sources of generation of $\hat{\mathbf{I}}^{(j)}_m$.
>
> **A6:** Perhaps, the statement ''$\hat{\mathbf{I}}^{(j)}_m$ is generated from $\hat{\mathbf{z}}^{(j)}_c$'' makes you confused. In fact, there is only one way to achieve $\hat{\mathbf{I}}^{(j)}_m$. As shown in Algorithm 1, $\hat{\mathbf{I}}^{(j)}_o$ is generated from $\mathbf{z}_m$ and $\hat{\mathbf{z}}^{(j)}_c$. That $\hat{\mathbf{I}}^{(j)}_m$ is obtained from $\hat{\mathbf{I}}^{(j)}_o$ with an image mask $\mathbf{M}$, i.e., $\hat{\mathbf{I}}^{(j)}_m=\hat{\mathbf{I}}^{(j)}_o\odot\mathbf{M}$ (refer to Figure 2), where $\odot$ denotes the Hadamard product.
>
>  > **Q7:** The acronym ''PGM'' (Line 69) is not introduced previously.
>
> **A7:** ''PGM'' is the abbreviation of the probabilistic graph model. We add the acronym ''PGM'' in Line 45.
>
> ----
> [1] Diederik P Kingma and Max Welling. Auto-Encoding Variational Bayes. ICLR, 2014.
> [2] Adji B. Dieng et al. Avoiding Latent Variable Collapse With Generative Skip Models. AISTATS, 2019.
> [3] Casper Kaae Sønderby et al. Ladder Variational Autoencoders. NeurIPS, 2016.
> [4] Leon A. Gatys et al. A Neural Algorithm of Artistic Style. arxiv 1508.06576.
> [5] Anders Boesen Lindbo Larsen et al. Autoencoding beyond pixels using a learned similarity metric. ICML, 2016.

---

> ### Author Response · Authors · 2022-08-06
> **Further Discussion**
>
> Dear Reviewer Nbcm:
>
> Thanks a lot for your valuable comments to enhance this paper. Are there unclear explanations here based on our response? We could further clarify them.
>
> Best,
> Authors

---

### Official Review · Reviewer_Dmgt · 2022-07-11

**Rating:** 6
**Confidence:** 3
**Soundness:** 3 good
**Presentation:** 2 fair
**Contribution:** 3 good

**Summary:**

For pluralistic image completion, the paper presents a probabilistic method that introduces a Gaussian mixture model(GMM) to build the interaction of some variables in image completion procedure. The GMM whose parameters are task-related is beneficial to the diversity and reliability of completion results. The method is evaluated on five datasets, i.e., CelebA-HQ, FFHQ, Paris StreetView, Places2, and ImageNet.

**Questions:**

Although I think this paper is an overall interesting paper, and most the experimental results are effective, I still have some concerns about this paper (more details could be found in weakness):

-the motivation  is a bit hard to follow;

-some more comparison to the related works is required;

-some details of evaluation are also required;

-the writting also could be improved.

**Limitations:**

The authors claimed that the limitation is described in the checklist, however, I didn't find the explicit discussion about the limitation. I find some comparison to ICT, which obtains better results than the proposed method, due to the using the larger model with transformer architecture.

**Strengths And Weaknesses:**

Strengths:

+The probabilistic method is interesting, its theoretical derivation and implementation steps are easy to follow.

+The proposed method is effective on the experimental datasets.

Weakness:

-The motivation for this paper is a bit hard to follow (in Line 5-7 and 32-41). Specifically, what is the constraints for visual reality? What does the word "task-agnostic" mean,  in this particular scenario, what is the specific meaning of "task" in this term? It's better to explain the in more details.

-The paper lacks more discussions about the correlation and differences between its own method PICME and existing works for pluralistic image completion in Line 110-114. Meanwhile, the experimental tables lacks the comparison results of some cited methods, such as [39,25], etc.

-About the Mixture-of-Experts Model, the size of the number of primitives k is not specifically discussed and analyzed, so the difference in effectiveness between the single Gaussian model when k=1 and the GMM when k>1 cannot be verified.

-The paper has some typos and should be checked carefully, such as the lack of a predicate in Line112 and lacking quoting symbols in the experimental tables.

---

> ### Author Response · Authors · 2022-08-02
> **Response to Reviewer Dmgt**
>
> Thanks for your comments. We address your concerns as follows.
>
> > **Q1:** The motivation is a bit hard to follow. More details are needed.
>
> **A1:** We address your concerns in the following aspects:
> - The constraints for visual reality mean that, in pluralistic image completion, we need to ensure the reality of completed images. For example, regarding human face images, we should make the completed images close to real faces, rather than random completion. For this goal, we need the constraints for visual reality, e.g., reconstruction loss and perception loss.
> - The term ''task'' refers to a specific scenario of pluralistic image completion. For example, in a certain task, we need to finish the image completion given a fixed dataset. The term ''task-agnostic'' means that, existing works push the diversity of completed images with prior knowledge, e.g., a predefined hyper-parameter. The knowledge is not related to the certain task and this given dataset.
>
> > **Q2:** Some more comparison to the related works is required.
>
> **A2:** Thanks. We find that some mentioned related works [25,39] do not provide source codes for the reproduction of results. We here add the baselines VQ-VAE [1] and AOT-GAN [2] for comparison. The experiments are conducted on the dataset Places2 with random masking. We provide the results in Table 3, which verify the effectiveness of our method. The results are added in Appendix F.
>
> | Method  | PSNR $\uparrow$ | SSIM $\uparrow$ | MAE $\downarrow$ | FID $\downarrow$ | LPIPS $\uparrow$ | DIV-FID $\uparrow$ |
> | ------- | --------------- | --------------- | ---------------- | ---------------- | ---------------- | ------------------ |
> | VQ-VAE  | 24.031          | 0.819           | 0.0309           | 30.747           | 0.083            | 23.676             |
> | AOT-GAN | 25.455          | 0.831           | 0.0233           | 26.427           | 0.089            | 25.843             |
> | PICME   | **25.467**      | **0.838**       | **0.0223**       | **24.334**       | **0.091**        | **28.033**         |
>
> Table 3 The results of comparing our PICME with VQ-VAE and AOT-GAN.
>
> > **Q3:** The details of evaluations with respect to the number of primitives $k$ are also required.
>
> **A3:** We supplement the experiments with different number of primitives, i.e., $k=1,2,3,4,5$. The experiments are conducted on the dataset CelebA-HQ using center masking. For every possible $k$, fifty images are generated to compare. We provide experimental results in Table 4. The results show that when we choose a larger $k$, our model is able to maintain quantitative metrics while increasing the diversity of images. The results are added in Appendix G.
>
> | $k$    | PSNR $\uparrow$ | SSIM $\uparrow$ | MAE $\downarrow$ | FID $\downarrow$ | LPIPS $\uparrow$ | DIV-FID $\uparrow$ |
> | ---- | --------------- | --------------- | ---------------- | ---------------- | ---------------- | ------------------ |
> | 1    | 26.792          | 0.961           | 0.0201           | 12.011           | 0.084            | 24.223             |
> | 2    | 26.979          | 0.965           | 0.0198           | 11.579           | 0.089            | 26.001             |
> | 3    | 27.152          | 0.960           | 0.0197           | 11.362           | 0.092            | 27.314             |
> | 4    | 26.943          | 0.964           | 0.0197           | 11.638           | 0.096            | 27.540             |
> | 5    | 27.121          | 0.966           | 0.0195           | 11.157            | 0.098            | 27.756             |
>
> Table 4 Evaluations of the number of primitives $k$.
>
> > **Q4:** The writing could be improved.
>
> **A4:** Thanks for your comments. In the rebuttal revision, We have revised the sentence in Line 112 and added quoting symbols in the experimental tables.
>
> > **Q5:** The explicit discussion about the limitation.
>
> **A5:** Thanks for mentioning this point. The limitation is that our PICME does not outperform ICT sometimes with respect to common metrics, since ICT uses a much larger transformer-based structure and iterative Gibbs sampling, which make ICT's inference slow. We discuss this limitation explicitly and more clearly in the revised version of the main paper.
>
> ----
> [1] Jialun Peng et al. Generating Diverse Structure for Image Inpainting With Hierarchical VQ-VAE. CVPR, 2021.
> [2] Yanhong Zeng et al. Aggregated Contextual Transformations for High-Resolution Image Inpainting. IEEE Transactions on Visualization and Computer Graphics, 2022.

---

> ### Author Response · Authors · 2022-08-06
> **Further Discussion**
>
> Dear Reviewer Dmgt:
>
> Thanks a lot for your insightful comments. We tried our best to address your concerns. Are there unclear explanations here? We could further clarify them.
>
> Best,
> Authors

---

### Official Review · Reviewer_cWnv · 2022-07-12

**Rating:** 6
**Confidence:** 2
**Soundness:** 3 good
**Presentation:** 3 good
**Contribution:** 3 good

**Summary:**

This paper proposes an end-to-end probabilistic method. The authors introduce a unified probabilistic graph model that represents the complex interactions in image completion. The entire procedure of image completion is then mathematically divided into several sub-procedures, which helps efficient enforcement of constraints. The sub-procedure directly related to pluralistic results is identified, where the interaction is established by a Gaussian mixture model (GMM). The inherent parameters of GMM are task-related, which are optimized adaptively during training, while the number of its primitives can control the diversity of results conveniently. The proposed method reaches state-of-the-art performance on various datasets.

**Questions:**

See above.

**Ethics Review Area:**

["I don’t know"]

**Limitations:**

See above.

**Strengths And Weaknesses:**

Strengths:
- The theoretical proof is well conducted.
- Experiments are convincing


Weakness:
- The shown figures are mostly low-resolution. Can the proposed method generate high-resolution images?
- Lack of comparison with other generative models. E.g., GAN/VAE-based or diffusion models.
- What is the advantages of using Mixture-of-Experts to do variational inference compared to traditional variational models (e.g., VAE)?

---

> ### Author Response · Authors · 2022-08-02
> **Response to Reviewer cWnv**
>
> Thanks for your comments. We address your concerns as follows.
>
> > **Q1:** Can the proposed method generate high-resolution images?
>
> **A1:** Yes, it can. The experiments in this work are before conducted with the image size $256\times256$. To show that our method can generate higher-resolution images, we set the image size to $512\times512$. Experiments are performed on the datasets FFHQ. We have supplemented the results in Appendix H of the rebuttal-revision version.
>
> > **Q2:** Lack of comparison with other generative models.
>
> **A2:** We address your concern from the following aspects:
> - Pluralistic image completion is a realistic but rather challenging problem. Only a few generative models target this problem. Therefore, most of generative models cannot act as baselines for a fair comparison.  In this work, we actually have exploited VAE-based generative models for comparison, e.g., PIC.
> - To make experiments more comprehensive, we compare our PICME with VQ-VAE [1] and AOT-GAN [2] on Places2. Random masking is used. The results are provided in Table 2, which demonstrate the effectiveness of our method.
>
>
> | Method  | PSNR $\uparrow$ | SSIM $\uparrow$ | MAE $\downarrow$ | FID $\downarrow$ | LPIPS $\uparrow$ | DIV-FID $\uparrow$ |
> | ------- | --------------- | --------------- | ---------------- | ---------------- | ---------------- | ------------------ |
> | VQ-VAE  | 24.031          | 0.819           | 0.0309           | 30.747           | 0.083            | 23.676             |
> | AOT-GAN | 25.455          | 0.831           | 0.0233           | 26.427           | 0.089            | 25.843             |
> | PICME   | **25.467**      | **0.838**       | **0.0223**       | **24.334**       | **0.091**        | **28.033**         |
>
> Table 2 The results of comparing our PICME with VQ-VAE and AOT-GAN.
>
> > **Q3:** The advantages of using Mixture-of-Experts to do variational inference compared to traditional variational models (e.g., VAE).
>
> **A3:** The advantages of using Mixture-of-Experts in this paper can be summarized as follows:
> - Mixture-of-Experts have larger capacities to satisfy different patterns in images. Hence, the use of Mixture-of-Experts in pluralistic image completion can promote the diversity of generated images. Experimental results in Table 3 of the main paper support our claim well.
> - We design a new objective function based on the frequency loss. The objective makes the inherent parameters of Mixture-of-Experts for diversity are task-related. Compared with traditional variational models, the advantage makes our method produce more reasonable results. Both qualitative and quantitative comparisons with state-of-the-art methods demonstrate the superiority of our method.
>
> ----
> [1] Jialun Peng et al. Generating Diverse Structure for Image Inpainting With Hierarchical VQ-VAE. CVPR, 2021.
> [2] Yanhong Zeng et al. Aggregated Contextual Transformations for High-Resolution Image Inpainting. IEEE Transactions on Visualization and Computer Graphics, 2022.

---

> > ### Comment · Reviewer_cWnv · 2022-08-09
> > **Response to the Rebuttal**
> >
> > Thank you for your clarification. I still have some concerns regarding performance and motivation.
> >
> > 1. I've checked the results in Appendix. H. It seems like the visual quality is less satisfactory. Especially, the boundary of bounding box contains some obvious artifacts.
> > 2. Can you share more insight about why Mixture-of-Experts have larger capacities? Also, I share the same concern with Reviewer oAsX that MoE might not be a suitable term and directly use the `Gaussian mixture model` would be better. This is important to me and related to the acceptance of this submission.
> >
> > Best,

---

> > > ### Author Response · Authors · 2022-08-09
> > > **Thanks for Your Feedback**
> > >
> > > Thanks for your response. We address your concerns one by one.
> > >
> > > > **Q1:** On the generated higher-resolution images.
> > >
> > > **A1:** Generally speaking, in low-level vision, to generate higher-resolution images, we need higher-capacity deep networks and more iterations. Other settings (e.g., the learning rate and optimizer) are different sometimes. Due to the limited time for rebuttal and in order to present the results as quickly as possible, we just adjusted the fully-connected layer of the original network so that it can fit the image size of 512$\times$512. Other settings are not changed. Therefore, the generated quality is not as excellent as in the cases of the image size 256$\times$256.
> > >
> > > Thanks for your suggestions. Our approach is not limited to a specific network structure. Because the revision deadline is too close, we promise to use a larger deep network and more iterations to generate higher-resolution images.
> > >
> > > > **Q2:** Why Mixture-of-Experts have larger capacities?
> > >
> > > **A2:** We provide our response below.
> > > - The larger capacities of Mixture-of-Experts have been claimed in Section 2.3.9 "Mixtures of Gaussians" of [1]. That is, "by using a sufficient number of Gaussians, and by adjusting their means and covariances as well as the coefficients in the linear combination, almost any continuous density can be approximated to arbitrary accuracy", while "a simple Gaussian distribution is unable to capture the complex structure" of data, especially the non-unimodal distributions.
> > > - For more insight into this claim, we strongly recommend an intuitive illustration of why ''Mixture-of-Experts'' have larger capacities in Section 2.3.9 "Mixtures of Gaussians" [1].
> > > - Taking the data distribution in Figure 2.21 of [1] for an example, where the true distribution of data is made by two clumps of data samples, the simple Gaussian distribution cannot capture this structure and indeed places much of its probability mass in the central region between the clumps, while the GMM can fit the distribution well.
> > >
> > > > **Q3:** It is better to use the Gaussian mixture model directly.
> > >
> > > **A3:** Thanks for your advice. We will use the Gaussian mixture model directly. In the rebuttal stage, we cannot modify the paper title in the submission system. At this stage, for consistency, we revise the name of Section 3.3 to ''Gaussian Mixture Model''. We will finish all modifications in the final version.
> > >
> > > ----
> > > [1] Christopher M. Bishop. Pattern Recognition and Machine Learning. Springer. URL: http://users.isr.ist.utl.pt/~wurmd/Livros/school/Bishop%20-%20Pattern%20Recognition%20And%20Machine%20Learning%20-%20Springer%20%202006.pdf

---

> > > > ### Comment · Reviewer_cWnv · 2022-08-09
> > > > **Response**
> > > >
> > > > Thanks for the clarification, my concerns have been addressed. I'll change my rating. I understand that the limited rebuttal period may not enable enough resources for training high-res images and I encourage authors to prepare better quality results in the final version upon acceptance.

---

> > > > > ### Author Response · Authors · 2022-08-09
> > > > > **Response**
> > > > >
> > > > > Thanks! We will prepare better quality results in the final version!

---

> ### Author Response · Authors · 2022-08-06
> **Further Discussion**
>
> Dear Reviewer cWnv:
>
> Thanks again for your efforts in reviewing. Would you mind checking our response and confirming if there are unclear explanations?
>
> Best,
> Authors

---

### Official Review · Reviewer_oAsX · 2022-07-13

**Rating:** 6
**Confidence:** 3
**Soundness:** 2 fair
**Presentation:** 2 fair
**Contribution:** 2 fair

**Summary:**

Problem: The paper deals with the problem of pluralistic image completion. This means that we want to generate multiple possible complete images.

Methodology: The authors propose a method which uses probabilistic graph models for the problem. This allows to handle visual reality and diversity explicitly by adding constraints and using a Gaussian mixture model (GMM).

Experiments: The proposed method produces good(most often state-of-the-art) results on various datasets like CelebA-hQ, Places2 and ImageNet both in qualitative and quantitative terms.

########## POST REBUTTAL ###
I have updated my rating as the authors have addressed my 2 key concerns.

**Questions:**

1. Please don't use the phrase 'Mixture of experts' because mixture of experts means something else in deep learning these days. It refers to models like LiMoE and Switch Transformer[B]. I don't see why you cannot just use GMM because that is all you are using. Can the authors please give a justification as to why this term is used?
2. Ablation studies: Can the authors provide some ablation studies that will help use understand the contribution of the different components? For instance, the effect of the different losses used.


[A] LIMoE: Learning Multiple Modalities with One Sparse Mixture-of-Experts Model

[B] Switch Transformers: Scaling to Trillion Parameter Models with Simple and Efficient Sparsity
William Fedus, Barret Zoph, Noam Shazeer
https://arxiv.org/abs/2101.03961



**Limitations:**

No, not at all. I am quite surprised that a GAN paper is not discussing negative impacts despite all the deep fakes that have been propagation around and bringing bad publicity to AI.

**Strengths And Weaknesses:**

Strengths:
1. Problem Relevance: The problem is quite relevant because diverse generation is more important than unique generation.
2. Quality: I like the way the method section has been structured. Starting from the Objective ( sec 3.2) is a great idea. Because this allows future work to also build up on the objective and come up with different ways to solve it.
3. Clarity: The paper is well written and clear.

Weaknesses:
1. Missing ablation studies

---

> ### Author Response · Authors · 2022-08-02
> **Response to Reviewer oAsX**
>
> Thanks for your comments. We address your concerns as follows.
>
> > **Q1:** The justification as to why the term ''Mixture of experts'' (MoE) is used.
>
> **A1:** Although the works, especially the big sparse MoE, you mentioned are famous nowadays, we think our design is still a kind of the MoE structure under the definition proposed in [1]. In more detail, [1] calls the structure with experts and a gating function switching the expert as the MoE structure. In our design, we consider the component Gaussians as experts and the Categorical distribution as the gating function. Therefore, the term ''Mixture of experts'' is used in this paper.
>
> Thanks for your nice comments. We cite the reference you mentioned and add the discussion about the use of the term ''Mixture-of-Experts'' in Appendix I. Also, this term may change to ''Mixture Distribution'' if there is no further suggestion.
>
> > **Q2:** Add ablation studies to help understand the contribution of the different components. For instance, the effect of the different losses used.
>
> **A2:** Thanks for your suggestion. We provide ablation studies, where experiments are conducted on CelebA-HQ with the center masking and $k=6$. Note that the drop of the loss $L_\text{R}$ will make the image completion unsupervised, resulting in rather unrealistic completed images. Therefore, the loss $L_\text{R}$ is added by default. The experimental results are provided in Table 1, which demonstrate the effectiveness of different components. We also add the results in Appendix E of the revised paper.
>
> |    $L_\text{F}$     |   $L_\text{BM}$   |  $L_\text{ELBO}$  |    $L_\text{A}$     | PSNR $\uparrow$ | SSIM $\uparrow$ | MAE $\downarrow$ | FID $\downarrow$ | LPIPS $\uparrow$ | DIV-FID $\uparrow$ |
> | :----------: | :----------: | :----------: | :----------: | --------------- | --------------- | ---------------- | ---------------- | ---------------- | ------------------ |
> | $\checkmark$ | $\checkmark$ | $\checkmark$ | $\checkmark$ | 27.098          | 0.964           | 0.0196           | 11.021           | 0.098            | 27.923             |
> | $\checkmark$ | $\checkmark$ | $\checkmark$ |     $-$      | 27.034          | 0.961           | 0.0194           | 11.193           | 0.095            | 27.534             |
> | $\checkmark$ | $\checkmark$ |     $-$      |     $-$      | 27.035          | 0.961           | 0.0198           | 11.202           | 0.094            | 27.068             |
> | $\checkmark$ |     $-$      |     $-$      |     $-$      | 26.251          | 0.922           | 0.0232           | 11.589           | 0.086            | 26.238             |
>
> Table 1 Ablation study of PICME.
>
> > **Q3:** Discussions on possible negative impacts.
>
> **A3:** Thanks for mentioning this point. We discuss the possible negative impacts of this paper, as concerned in many methods based on generative models. That is, generated images with our method may danger to the security of AI systems, in the form of deep fakes. We add the discussion in Appendix J, due to the limited page of the main paper.
>
> ----
> [1] Robert Jacobs et al. Adaptive Mixtures of Local Experts. Neural Computation, 1991.

---

> ### Author Response · Authors · 2022-08-06
> **Further Discussion**
>
> Dear Reviewer oAsX:
>
> Thanks a lot for your efforts in reviewing this paper. We tried our best to address the mentioned concerns. Are there unclear explanations here? We could further clarify them.
>
> Best,
> Authors

---

### Author Response · Authors · 2022-08-02
**General Response**

We thank the reviewers for their insightful and constructive reviews of our manuscript. We are glad for the positive ratings from all reviewers. In addition, we are highly encouraged to hear that the reviewers found that the core idea is interesting and contributing (Reviewers oAsX, Dmgt, and Nbcm), the theoretical analysis is solid (Reviewers cWnv, Dmgt, and Nbcm), and experiments are overall convincing (Reviewers cWnv and Dmgt). Based on the reviews, we provide a general response here to the points raised by multiple reviewers, and individual responses below to address each reviewer’s concerns. Importantly,

(1) Regarding questions about experiments, we have addressed the concern as follows:
- For Reviewer oAsX, we add ablation studies to help understand the contribution of the different components in our method.
- For Reviewer cWnv, we add experiments to show that our method can generate high-resolution images effectively.
- For Reviewer Dmgt, we compare our method with two related works as suggested, and add the evaluations w.r.t. the number of primitives $k$.

(2) Regarding questions about the idea and technical details, we have addressed the concern as follows:
- For Reviewer oAsX, we justify the use of MoE.
- For Reviewer Dmgt, we add details to explain the motivation.
- For Reviewer Nbcm, we provide more descriptions of ''1-step Monte Carlo'' and more explanations for our method.

We have revised our draft according to your comments. Major revisions are highlighted in blue. We sincerely thank all the reviewers. Please feel free to let us know if further details/explanations would be helpful.

Best,
Authors

---

### Author Response · Authors · 2022-08-08
**Further Discussions**

Dear reviewers:

Thanks again for your efforts in reviewing. The discussion deadline is approaching. Are there unclear explanations here? We could further clarify them.

Best,
Authors

---

> ### Comment · Area_Chair_D2UB · 2022-08-08
> **End discussion period**
>
> Dear Reviewers
>
> It would also be interested if you could provide feedback on the rebuttal and each other’s reviews, and possibly update your score. The discussion period ends tomorrow ! So please ask any remaining clarifications to the authors ASAP.
>
> best regards
> AC

---

> > ### Author Response · Authors · 2022-08-09
> > **Further Discussions**
> >
> > Dear Reviewers:
> >
> > Thanks again for your efforts in reviewing. We are still looking forward to your reply. Would you mind checking our response and confirming if there are unclear explanations?
> >
> > Best,
> > Authors

---

### Meta-Review · Area_Chair_D2UB · 2022-08-25

**Recommendation:** Accept
**Confidence:** Certain

**Metareview:**

The paper addresses the pluralistic image completion problem. Initial reviews were borderline accept (2x) , weak accept (2x). The authors provided a rebuttal. Both borderline reviewers upgraded to weak accept. The AC agrees and considers the paper a solid contribution to NeurIPS and recommends acceptance.

**Award:**

No

---

### Decision · Program_Chairs · 2022-09-14

Accept